# Role of endoplasmic reticulum stress in impaired neonatal lung growth and bronchopulmonary dysplasia

Kirkwood A. Pritchard, Jr.[1,2], Xigang Jing[2,3], Michelle Teng[3], Clive Wells[4], Shuang Jia[3], Adeleye J. Afolayan[2,3], Jason Jarzembowski[2,5], Billy W. Day[6], Stephen Naylor[6], Martin J. Hessner[2,3], G. Ganesh Konduri[2,3], Ru-Jeng Teng[2,3]*

1 Division of Pediatric Surgery, Department of Surgery, Medical College of Wisconsin, Wauwatosa, Wisconsin, United States of America, 2 Children's Research Institute, Medical College of Wisconsin, Wauwatosa, Wisconsin, United States of America, 3 Department of Pediatrics, Medical College of Wisconsin, Wauwatosa, Wisconsin, United States of America, 4 Electron Microscope Facility, Medical College of Wisconsin, Wauwatosa, Wisconsin, United States of America, 5 Division of Pediatric Pathology, Department of Pathology, Medical College of Wisconsin, Wauwatosa, Wisconsin, United States of America, 6 ReNeuroGen L.L.C. Milwaukee, Elm Grove, Wisconsin, United States of America

* rteng@mcw.edu

**Data Availability Statement:** All relevant data are located at the Gene Expression Omnibus Database under the accession numbers GSE197403 and GSM5916410-GSM5916420.

## Abstract

Myeloperoxidase (MPO), oxidative stress (OS), and endoplasmic reticulum (ER) stress are increased in the lungs of rat pups raised in hyperoxia, an established model of bronchopulmonary dysplasia (BPD). However, the relationship between OS, MPO, and ER stress has not been examined in hyperoxia rat pups. We treated Sprague-Dawley rat pups with tunicamycin or hyperoxia to determine this relationship. ER stress was detected using immunofluorescence, transcriptomic, proteomic, and electron microscopic analyses. Immunofluorescence observed increased ER stress in the lungs of hyperoxic rat BPD and human BPD. Proteomic and morphometric studies showed that tunicamycin directly increased ER stress of rat lungs and decreased lung complexity with a BPD phenotype. Previously, we showed that hyperoxia initiates a cycle of destruction that we hypothesized starts from increasing OS through MPO accumulation and then increases ER stress to cause BPD. To inhibit ER stress, we used tauroursodeoxycholic acid (TUDCA), a molecular chaperone. To break the cycle of destruction and reduce OS and MPO, we used N-acetyl-lysyltyrosylcysteine amide (KYC). The fact that TUDCA improved lung complexity in tunicamycin- and hyperoxia-treated rat pups supports the idea that ER stress plays a causal role in BPD. Additional support comes from data showing TUDCA decreased lung myeloid cells and MPO levels in the lungs of tunicamycin- and hyperoxia-treated rat pups. These data link OS and MPO to ER stress in the mechanisms mediating BPD. KYC's inhibition of ER stress in the tunicamycin-treated rat pup's lung provides additional support for the idea that MPO-induced ER stress plays a causal role in the BPD phenotype. ER stress appears to expand our proposed **cycle of destruction**. Our results suggest ER stress evolves from OS and MPO to increase neonatal lung injury and impair growth and development. The encouraging effect of TUDCA indicates that this compound has the potential for treating BPD.

**Funding:** Children's Research Institute - Program Support, Research Unit Leader, and HL128371 (KAP); Children's Wisconsin Foundation, American Diabetes Association 1-19-ICTS-129, R01DK125014, and R01DK121528 (MJH); HL136597 and HL144519 (GGK); Children's Research Institute Pilot Innovative Research Grant and Department of Pediatrics Internal Support (RJT).

**Competing interests:** Kirkwood A. Pritchard Jr., Ph.D., and Stephen Naylor, Ph.D. are co-founders and co-owners, and Billy W. Day is a co-owner of ReNeuroGen L.L.C., an early-stage virtual pharmaceutical company that is currently developing KYC as a treatment for stroke, multiple sclerosis, vasculopathy in sickle cell disease, and evaluating the potential use of KYC in the treatment of BPD. All other authors have no conflicts, financial or otherwise. This does not alter our adherence to PLOS ONE policies on sharing data and materials.

## Introduction

Bronchopulmonary dysplasia (BPD) is a common complication of preterm birth that affects about 50% of all extremely premature infants [1]. More than 15,000 infants are diagnosed with BPD each year in the USA [2]. BPD survivors frequently suffer from exercise intolerance or suboptimal lung function that persists long into adulthood [3]. The BPD phenotype is highly variable, with different types of oxidative stress (OS), lung injury, and repair [4]. These complexities have hampered detailed mechanistic studies of BPD, resulting in a limited number of effective therapies for BPD [5]. For example, landmark studies of gentle ventilation, judicious use of oxygen, ante- and postnatal steroid treatment, early caffeine administration, optimized nutritional support, and postnatal vitamin A injection have been shown to decrease some aspects of BPD [6]. Yet, the annual number of BPD cases has not changed significantly over the past several decades [2].

Premature infants often receive supplemental oxygen and mechanical ventilation to maintain their metabolism. Paradoxically, these very interventions and infections are the major contributors to BPD onset and progression [2]. Both types of respiratory support have been reported to increase OS and injury to the immature lungs of neonates who cannot mount an appropriate antioxidant response to protect their lungs [7]. The relationship between OS and inflammation in neonates treated with supplemental oxygen is well established [8].

Animal models facilitate our understanding of the pathophysiology of BPD. There is no perfect animal model that can imitate premature human neonates. Rats are born at the saccular stage, reminiscent of premature infants at 24–28 weeks of gestation. The alveolar formation starts at postnatal day 4 (P4) and persists to P21 in rats, corresponding to premature infants at 36 weeks of gestation and children at 2–6 years. Rat pups are commonly used to study BPD because when they are exposed to more than 80% oxygen before P4 they develop a lung phenotype with alveolar simplification [9]. The major limitation of using the rat model is that rat pups are born with a mature antioxidant system, so higher oxygen concentrations are needed to induce the BPD phenotype. The benefits of using the rat model are a short lifespan, heterogeneous genetics of outbred strains mimicking the human condition, and a high number of offspring that can be randomized for parallel studies. We have used this rat model to study BPD mechanisms for several years [10, 13]. Recently we reported that chronic hyperoxia (HOX) in neonatal rat pups causes a BPD phenotype by inducing a cycle of destruction initiated by myeloperoxidase (MPO) and propagated by high mobility group box-1 (HMGB1) [10]. The temporal nature of this cycle of destruction suggests that the OS that occurs at the onset is *different* from the OS that develops as BPD progresses.

Previously it was reported that HOX and interferon-gamma induce lung injury neutrophil arrival by increasing endoplasmic reticulum (ER) stress in the lungs of neonatal mice [11]. Tong has shown that chronic HOX induces ER stress in neonatal rat lungs [12] by a mechanism that we have shown can be inhibited by administering caffeine early on [13]. As the OS generated by the cycle of destruction [10] appears in conflict with caffeine-inhibitable ER stress, additional studies are required to understand how OS induces ER stress and the BPD phenotype as the disease progresses. We thus hypothesize here that ER stress propagates the cycle of destruction initiated by OS and MPO to cause BPD, and as such inhibiting ER stress should protect neonatal lungs against HOX injury. In this manuscript, we evaluate the role of ER stress in BPD as well as the combined contributions of MPO, and OS-mediated ER stress in BPD. To investigate OS and inflammation induced by the cycle of destruction we treated tunicamycin (Tun-) and HOX-treated neonatal rat pups with N-acetyl-lysyltyrosylcysteine amide (KYC). This bioengineered tripeptide inhibits MPO from producing toxic oxidants such as hypochlorous acid [14]. To induce ER stress specifically and evaluate its role in BPD,

we treated neonatal rat pups with Tun. This mixture of nucleoside antibiotics inhibits protein post-translational N-glycosylation, impairing protein folding to induce the unfolded protein response (UPR) [15]. Attenuation of ER stress was investigated using Tun- and HOX- neonatal rat pups treated with tauroursodeoxycholic acid (TUDCA). TUDCA is a chemical chaperone that supports chaperone activity by interacting with unfolded proteins in the ER [16].

Our goals were to evaluate the role of ER stress in BPD and, in addition, determine whether inhibiting ER stress is an effective strategy for reducing BPD. Our studies show that chronic HOX induces BPD by increasing ER stress and that Tun-induced ER stress increases myeloid cell recruitment and MPO protein in the lung. TUDCA effectively decreases ER stress, myeloid cell recruitment, and MPO protein in the Tun- and HOX-treated neonatal rat lungs. Interestingly, KYC also reduces ER stress and MPO protein in Tun-treated neonatal rat lungs. These data suggest that MPO and OS are interconnected with ER stress in BPD and that targeting ER stress is an effective strategy for reducing BPD regardless of how ER stress is induced.

## Materials and methods

The ER stress antibody kit (#9956S) contains mouse antibody for CHOP, rabbit antibodies for PERK, IRE1α, GRP78, aVEGFR2 (#2479), and thioredoxin 1 (TRX, # 2298) was from Cell Signaling (Beverly, MA). The mouse antibody for caspase-12 (GTX59923) and rabbit antibody for heme oxygenase 1 (HO1, GTX101147) were from GeneTex (Irvine, CA). Rabbit antibodies for mitofusin-1 and -2 (MFN-1/-2, sc-50330/sc-50331), B-cell lymphoma 2 (BCl-2, sc-492), BCL2 Associated X (Bax, sc-6236), myeloperoxidase (MPO) heavy chain (sc-33596), dynamin-related protein 1 (DRP-1, sc-32898), and phospho-PERK (P-PERK, sc-32577) were from Santa Cruz Biotechnology (Dallas, Texas). Rabbit antibodies for phospho-IRE1α (P-IRE1α, NB100-2323) and X-box binding protein 1 (XBP1, NB100-78403), and mouse antibody for activating transcription factor 6 (ATF6, NBP1-40256) were from Novus Biologicals (Littleton, CO). Rabbit antibodies for acyl-CoA synthetase 4 (ACSL4, LS-B5818) and glucose-regulated protein 75 (GRP75, LS-C312939) were from LifeSpan BioSciences (Seattle, WA). Mouse anti-HT1-56 antibody against human type 1 alveolar cells (AT1) and anti-RT-40 antibody against rat AT1 were from Terrace Biotech (San Francisco, CA). Mouse antibody for rat endothelial cell antigen-1 (RECA-1, ab22492) was from Abcam (Cambridge, MA). Rabbit antibody against glutathione-S-transferase (GST, MBS355205) was from MyBioSource (San Diego, California). In Situ Cell Death Detection POD Kit (#11684817910) was from Roche Applied Science (Indianapolis, IN). Tunicamycin (Tun, #11445) was from Cayman Chemical (Ann Arbor, Michigan) and was freshly dissolved in dimethyl sulfoxide (DMSO) before administration. Tauroursodeoxycholic acid (TUDCA) sodium salt (#580549) was from EMD Millipore (Darmstadt, Germany) and dissolved in phosphate buffer solution (PBS) right before injection. KYC was synthesized using Fmoc [N-(9-fluorenyl)methoxycarbonyl] chemistry and purified as an acetate salt by Biomatik USA, LLC (Wilmington, DE). TFA was reduced to < 0.01%, as we previously reported [9], before use in *in vivo* experiments. GeneChip™ Rat Genome 230 2.0 Array (#99506) was from Thermo Fisher Scientific (Santa Clara, CA).

### Human subjects

Human paraffinized lung tissues were from de-identified autopsies cataloged by our Pediatric Pathology (Children's Wisconsin [CW], Milwaukee, WI). BPD diagnosis was established by both clinical information and pathologic findings [2]. Lung tissues from infants who died from non-pulmonary diseases or injuries, of comparable age and sex were obtained as controls and matched for age and sex at a 1:1 ratio. All protocols used to obtain the autopsy lung tissues

were reviewed and approved by CW's Institutional Review Board while consent was waived according to the nature of the study.

## Animal care

Time-dated-pregnant Sprague-Dawley rats were obtained from Envigo (Madison, WI) and were acclimated to our animal facility for seven days. The Medical College of Wisconsin Institutional Animal Care and Use Committee approved the use of animals which complied with the National Institute of Health Guide for the Care and Use of Laboratory Animals (AUA00002268). Nursing dams were cared for in a 12-h dark-light cycle and had free access to chow and water. According to the study design, two or four pregnant rats were used for each experiment. Pups from 2–4 dams were mixed and randomly reassigned to each nursing dam by sex. The dams and pups were cared for in either >90% oxygen chamber (HOX) or room air (NOX) from P1 to P10 (S1 Fig). Oxygen concentrations were continuously monitored with an oxygen sensor (Reming Bioinstruments Co., Redfield, NY). Pups were caged with nursing dams while the dams were alternated between oxygen environments to diminish oxygen toxicity.

## Animal treatment

To induce ER stress, Tun was administered intraperitoneally (IP) via a 30G insulin syringe (Beckon Dickinson, New York, NY) to induce ER stress. The dose of Tun was predetermined through a series of preliminary trials (from 0.05 to 1.5 mg/kg/dose at P1 or P3 given once or twice) according to the literature [9, 17], and the single injection of 0.1 mg/kg at P3 was chosen for this study for acceptable survival rate. This injection time was determined because the first wave of rapid alveolar formation occurs between P4 and P10 in rats [16]. The dose of TUDCA (100 mg/kg/d) was estimated through the human equivalent dose equation [18], and the recommended dose (30 mg/kg/d) used in human infants [19], which was within the reported ranges (10–1,500 mg/kg) in animal studies [20, 21]. The dose of KYC (10 mg/kg daily) was chosen, as we previously reported [10]. TUDCA and KYC were dissolved in PBS for daily IP injection starting at P2 through P10 via a 30G insulin syringe. Pups were euthanized with carbon dioxide. The lungs and heart were removed en bloc. A small cut was created on the left atrium. Ice-cold saline was gently infused through the right ventricle to remove blood from the lung before the inflation for histology or snap-frozen in liquid nitrogen for protein studies. The Tun-treated pups were euthanized either at P5 to study proteomic changes in the lungs or at P7 for lung morphometric analyses since our preliminary studies showed most pups started to have decreased activity and appetite two days after injection [22]. HOX rat pups were euthanized at P10.

## Histology

For histology studies, the trachea was cannulated, and the lungs were inflated with 10% neutral buffered formalin at 20 cm-$H_2O$ (1.9 kPa) for 1 h. Lungs were removed with the trachea securely tied with surgical silk under a pressure of 20 cm-$H_2O$ and fixed in 10% neutral buffered formalin for 24 h before being embedded in paraffin. Lung paraffin sections (5 μm) were mounted on SuperFrost plus-coated slides (Denville Scientific, Metuchen, NJ). After deparaffinization, sections were stained with hematoxylin and eosin (H&E). Images, devoid of major bronchi and large blood vessels, were captured using an Olympus IX 51 microscope fitted with a digital camera.

Mean linear intercept (MLI) was used to estimate the volume-to-surface ratio of acinar airspaces while radial alveolar count (RAC) and secondary septa were used to investigate the

complexity of the alveolar structure [23]. MLI and RAC were obtained as we have reported [13]. For the measurement of secondary septa, elastin was stained with resorcin-fuchsin and Van Gieson's solution.

*In situ* TUNEL staining was done according to the manufacturer's instructions using DAB as the chromophore and counterstained with H&E. Apoptosis was quantified by the percentage of brown-stain nuclei divided by the total number of nuclei. Myeloid cells were stained with the MPO antibody. The average of three sections per pup and randomly chosen five counts per section (15 counts/pup) were used for statistical analyses.

P-PERK or P-IRE1α immunofluorescence (IF) was used as the biomarker for ER stress in paraffinized tissue sections. HT1-56 and RT-40 antibodies (1:100) were used to identify human and rat AT1 cells. RECA-1 antibody was used to identify rat endothelial cells. Lung sections (5 μm) were stained with the respective antibody (1:100) overnight at 4°C, then incubated with AlexaFluor-conjugated secondary antibody for one hour at room temperature, followed by counterstaining with DAPI for IF or incubated with HRP-conjugated secondary antibody for one hour at room temperature followed by diaminobenzidine (DAB) treatment for immunohistochemical (IHC) stain. The integrated signals were processed and quantified using color deconvolution with Fiji v1.53f software.

## Transmission electron microscopy

Heart and lungs were obtained en bloc after euthanasia and perfused with a mixture of 2% formaldehyde and 2.5% glutaraldehyde in 0.1 M sodium cacodylate buffer (pH 7.4). The mixture was gently infused through the right ventricle until no visible blood could be seen coming out of the left atrium. The perfused lungs were then immersed in 2.5% glutaraldehyde overnight. Sections of 60 nm thickness were obtained by ultramicrotome. Images of the pulmonary vascular endothelial cells (EC) and type 2 alveolar cells (AT2) were obtained by Hitachi 600 electron microscope provided by our core facility. Images were obtained by personnel blinded to study group assignments.

## Transcriptomics

RNA was extracted by the TRIzol method. To lyse tissue, minced lung tissue was mixed with TRIzol by vortex, then 300 μl chloroform was added into the tube to lyse the tissue by shaking. After sitting at room temperature for 3 min, the tubes were centrifuged at 12,000g for 15 min at 4°C. The top aqueous layer was transferred into a new tube with 500 μl chloroform and mixed by shaking. The tubes were centrifuged at 12,000g for 15 min at 4°C, and the top aqueous layer was transferred into a new tube followed by adding 2 μl of linearized acrylamide and 500μl of isopropanol to mix by shaking. After sitting at room temperature for 10 min, the tubes were centrifuged at 12,000g for 20 min at 4°C. The supernatant was carefully pipetted without disturbing the pellet. RNA pellets were washed with 1ml of 75% ethanol. After mixing by inverting several times, the tubes were centrifuged at 7,500g for 5 min at 4°C. The supernatant was carefully removed by pipet, and the pellet was dried by natural evaporation. The dried pellet was resuspended in DEPC $H_2O$ to obtain an RNA concentration of ~200 ng/3 μl. RNAs were labeled and hybridized to Affymetrix RG230 2.0 arrays as described [24]. Array images were quantified with Affymetrix Expression Console Software, then normalized and analyzed with Partek Genomic Suite (Partek Inc, St. Louis, MO, USA). Gene expression differences were evaluated by principal component analyses and by non-parametric rank product tests and the determination of false discovery rates (FDR) to investigate the rate of type I errors in multiple testing [25]. Ontological analyses were conducted with ToppCluster [26]. Genesis was used to perform hierarchical clustering and generate heatmaps [26]. Data files have been

deposited at The National Center for Biotechnology Information Gene Expression Omnibus
(GSE197403).

## Western blotting

Lung lysates were prepared by homogenizing in MOPS buffer (20 mM 3-*N*-morpholino-pro-
pane-sulfonic acid, 2 mM EGTA, 5 mM EDTA, 30 mM NaF, 10 mM β-glycerophosphate, 10
mM Na pyrophosphate, 2 mM Na orthovanadate, 1 mM PMSF, 0.5% NP-40, 1% protease
inhibitor cocktail, and 1% phosphatase inhibitor cocktails 2 and 3, pH 7.0) via Bullet Blender
(Next Advance, Inc., Averill Park, NY). Proteins were resolved by SDS-PAGE, transferred to
nitrocellulose membranes, and then probed with primary antibodies overnight at 4˚C for
immunoblotting. Signals were generated after incubation with horseradish peroxidase-conju-
gated goat anti-rabbit (1:4,000) or anti-mouse (1:4,000) IgG using Pierce™ ECL Western Blot-
ting Substrate (#32209, Thermo Scientific™, Rockford, IL) and recorded by iBright FL1000
Imaging System (Invitrogen™). Integrated optical density (IOD) was processed with ImageJ
software, and the β-actin signal was used as the loading control.

## Statistical analysis

Data were analyzed by MedCalc Statistical Software version 15.2.1 (MedCalc Software bvba,
Ostend, Belgium) or GraphPad Prism version 9.0.0 for Windows, GraphPad Software (San
Diego, CA). Unpaired and paired t-test or Mann Whitney U test was used to compare two
groups, while one-way ANOVA with Student-Newman-Keuls post-hoc test after Levene's test
for equality of error variances was used to compare data between more than two groups. The
difference in body weight among groups was performed using two-way ANOVA. Values are
expressed as mean ± SD, and a scatterplot was used for the figures. All transcriptomes with a
difference in expression level between two groups reaching p<0.05 and Benjamini-Hochberg
Step-Up FDR <0.1 were uploaded into ToppCluster for gene enrichment analysis [27]. All p-
value <0.05 were considered statistically significant.

## Results

### ER stress is increased in both human and hyperoxic rat lungs

There was no difference in the age of death between sex-matched human BPD and controls
(25~375 days; n = 10 for each group, p = 0.82; Wilcoxon signed-rank test). Characteristic pic-
tures including the thickened alveolar wall, decreased alveolar number, hyper-cellularity,
denuded cells occupying the alveolar sac, and inflammatory cell infiltration were seen in all
human BPD lungs. Immunofluorescent (IF) stains of P-IRE1α and P-PERK were used to
detect ER stress. Human BPD lungs had a significant co-localization of AT1 and P-IRE1α than
the control lungs (Fig 1A). Integrated fluorescence intensity confirmed that ER stress was sig-
nificantly increased in human BPD lungs (~1.8-fold). The co-localization of P-PERK fluores-
cence with AT1 in the hyperoxic rat lungs was higher than in the normoxic control lungs (Fig
1B). The results from the IF stain in both human and hyperoxic rat BPD corroborate and con-
firm our previous report showing increased ER stress in BPD rat lungs as determined by
immunoblots [13]. The transcriptomic study of rat lungs obtained at P10 revealed 3928 genes
significantly increased in expression with FDR <0.1 and 118 genes (out of 313 genes in
GO:0034976) annotated as ER stress-related indicating chronic HOX causes an upregulation
of ER stress-related genes (Fig 1C, S1 Fig, Table 1). Other upregulated genes included cell
death/apoptosis, ribosome biogenesis, oxidative phosphorylation, myeloid cell activation/
migration/degranulation, autophagy, et al. The gene set enrichment analysis (GSEA) identified

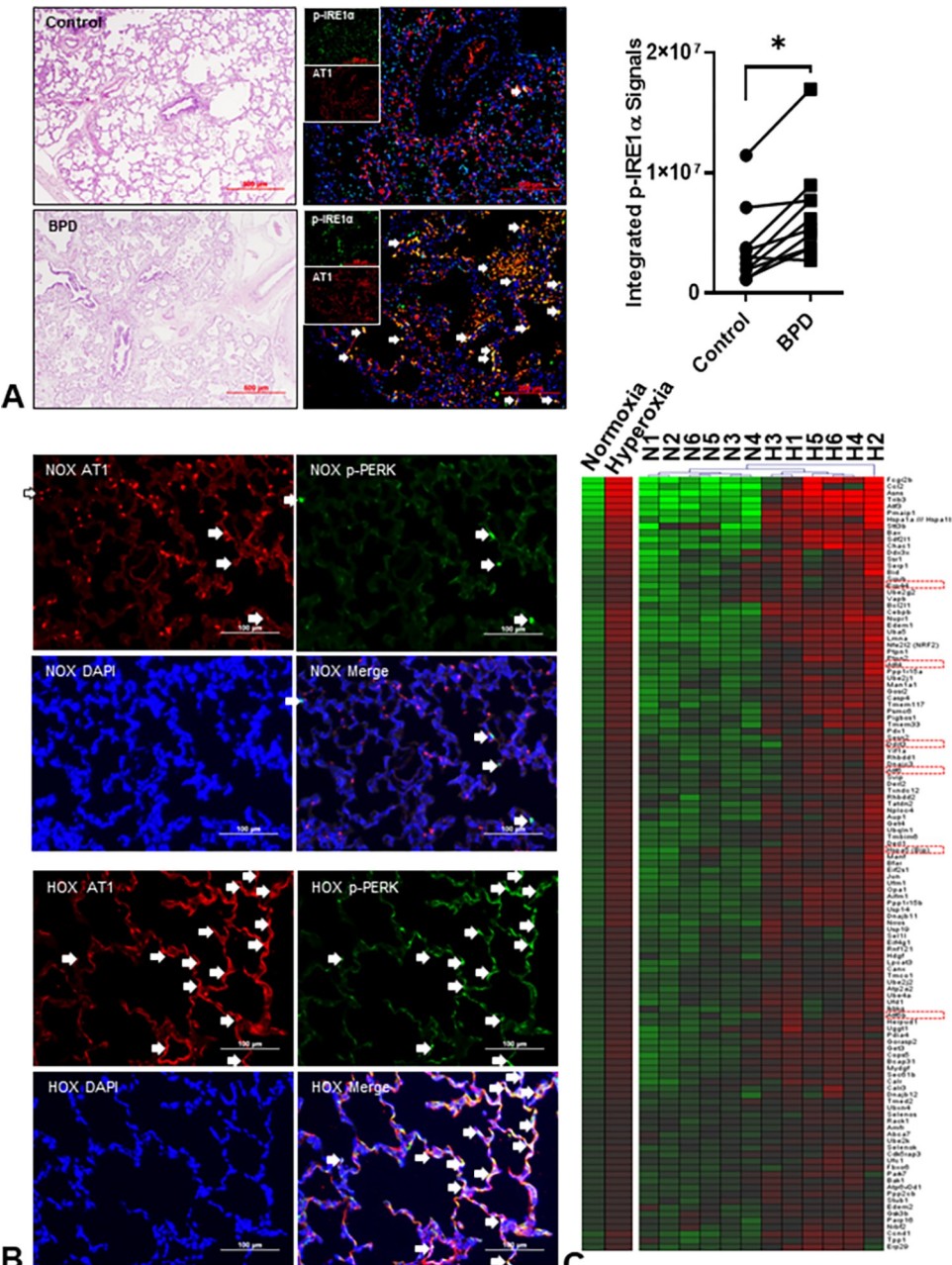

**Fig 1. Endoplasmic reticulum (ER) stress is increased in bronchopulmonary dysplasia (BPD).** (**A**) Representative images of human lungs from non-BPD and BPD infants. The BPD lungs have a simplified alveolar structure, thickened alveolar walls, denuded epithelial cells, and inflammatory cell infiltration. The increased co-localization of phospho-IRE1α (green) and type 1 alveolar epithelial cells (AT1, red) indicates increased ER stress in human BPD lungs. The integrated fluorescent density of colocalized (white arrows) phospho-IRE1α fluorescence and AT1 is significantly increased in BPD lungs ($2.7 \times 10^6$–$1.7 \times 10^7$ vs. $1.2 \times 10^6$–$1.2 \times 10^7$, n = 10, p = 0.0012 by Wilcoxon signed-rank test) than the age- and gender-matched (6 males and 4 females) control non-BPD lungs. (**B**) Phospho-PERK (green) and AT1 (red) were used to detect ER stress in rat lungs. The colocalization (white arrows) is markedly increased in HOX (hyperoxia, >90% $O_2$) BPD rat lungs at postnatal day 10 (P10). (**C**) The transcriptomic study is done by Affymetrix GeneChip™ Rat Genome 230 2.0 Array (n = 6 for both NOX and HOX; 3 males and 3 females in each group) on RNA obtained from lungs at P10. Gene expressions reaching significant differences (p<0.05) with FDR <0.1 are selected and uploaded into ToppGene for gene enrichment analysis and annotation per Gene Ontology Biological Processes. Totally 118 genes involved in response to ER stress (GO:0034976) are significantly upregulated in hyperoxic rat lungs, as shown in the heatmap. Markers commonly used to demonstrate an increased ER stress are highlighted in red squares. *: p<0.05.

**Table 1. Transcriptomic study reveals increased transcription of ER stress-related genes in hyperoxic BPD rat lungs compared to normoxic controls (n = 6) at P10.**

| Items | Biologic process | p-value | q-FDR Bonferroni | q-FDR B&H | q-FDR B&Y | Hit | Total genes |
|---|---|---|---|---|---|---|---|
| GO:0034976 | Response to ER stress | $1.391 \times 10^{-12}$ | $1.678 \times 10^{-8}$ | $7.840 \times 10^{-11}$ | $7.820 \times 10^{-10}$ | 118 | 313 |
| GO:0006986 | Response to unfolded protein | $1.446 \times 10^{-12}$ | $1.744 \times 10^{-8}$ | $8.110 \times 10^{-11}$ | $8.090 \times 10^{-10}$ | 83 | 194 |
| GO:0034620 | Cellular response to unfolded protein | $2.884 \times 10^{-9}$ | $3.477 \times 10^{-5}$ | $1.073 \times 10^{-7}$ | $1.070 \times 10^{-6}$ | 64 | 155 |
| GO:0030968 | Endoplasmic reticulum unfolded protein response | $7.003 \times 10^{-8}$ | $8.444 \times 10^{-4}$ | $2.035 \times 10^{-6}$ | $2.030 \times 10^{-5}$ | 54 | 132 |
| GO:1905897 | Regulation of response to ER stress | $1.737 \times 10^{-7}$ | $2.094 \times 10^{-3}$ | $4.781 \times 10^{-6}$ | $4.769 \times 10^{-5}$ | 40 | 89 |
| GO:0140467 | Integrated stress response signaling | $6.137 \times 10^{-7}$ | $7.400 \times 10^{-3}$ | $1.538 \times 10^{-5}$ | $1.535 \times 10^{-4}$ | 18 | 28 |
| GO:0030433 | Ubiquitin-dependent ERAD pathway | $4.645 \times 10^{-5}$ | $5.601 \times 10^{-1}$ | $8.071 \times 10^{-4}$ | $8.050 \times 10^{-3}$ | 33 | 83 |
| GO:0070059 | Intrinsic apoptotic signaling pathway in response to ER stress | $1.119 \times 10^{-4}$ | 1.000 | $1.804 \times 10^{-3}$ | $1.799 \times 10^{-2}$ | 28 | 69 |

ER: Endoplasmic reticulum; ERAD: Endoplasmic reticulum-associated protein degradation; UPR: unfolded protein response

NRF2-ARE (M39761, p = $2.82 \times 10^{-6}$) and NRF2 (M39454, p = $3.60 \times 10^{-6}$) pathways implying the lungs mounted a response to OS (S2 Fig). The relative transcript levels for NRF2, heme oxygenase 1 (HO1), SOD2, and thioredoxin reductase 1 (TRX) increased by 1.3-, 3.0-, 2.2-, and 1.9-fold (p<0.05, qFDR<0.05), respectively. Expression of GST, HO1, and TRX was increased as early as P4 (S3 Fig). Downregulated genes predominantly mediated cell proliferation, organ development/morphogenesis, and angiogenesis/vasculogenesis. (see original raw data at The National Center for Biotechnology Information Gene Expression Omnibus as GSE197403) (S1, S2 Tables).

## HOX rapidly increases ER stress in the lungs and impairs the survival rate and alveolar formation of neonatal rats

We previously reported that chronic HOX increases ER stress by P10 [13], but whether such changes occurred earlier is unknown. ER stress markers (p-PERK, p-IRE1α, cATF6, sXBP1, and CHOP) increased as early as P4 in HOX neonatal rat lungs, indicating that neonatal lungs are capable of mounting an ER stress response early on after HOX exposure (Fig 2A). As reported earlier, HOX once again significantly decreased survival rate and alveolar formation at P10 (Fig 2B and 2C) [13].

## Tun induces ER stress in neonatal rat lungs

One dose of Tun (0.1 mg/kg) at P3 significantly decreased the survival rate of rat pups and decreased the weight gain and activity of all surviving pups. Weight was significantly reduced at P5, two days after injection (Fig 3A). The expression of ER stress sensors (P-PERK, PERK, P-IRE1α, IRE1α, and cleaved ATF6) and downstream effectors (spliced XBP1 and CHOP) were all increased in the Tun-treated lungs as compared to the lungs of DMSO-treated control rat pups. The endogenous ER chaperone protein, GRP78, increased by 1.6-fold, conformed with increased spliced XBP1, and cleaved ATF6 to confirm the increased ER stress [28]. The dramatic decrease in N-glycosylated VEGFR2 (gVEGFR2) further supports that Tun directly disturbs ER function and explains why angiogenesis is suppressed (Fig 3B) [29]. The increased levels of CHOP and cleaved caspase-12 indicate ER stress-mediated apoptosis should be increased (Fig 3C), which was confirmed by the marked increase *in situ* TUNEL+ cells (Fig 3D). As expected, immunohistochemistry (IHC) for P-IRE1α was increased in Tun-treated rat lungs, especially inside the vasculature (Fig 3E). Immunofluorescent (IF) stain for P-PERK revealed its co-localization with vascular endothelial cells (Fig 3F). Both IHC and IF data confirm ER stress was increased in Tun-treated neonatal rat lungs.

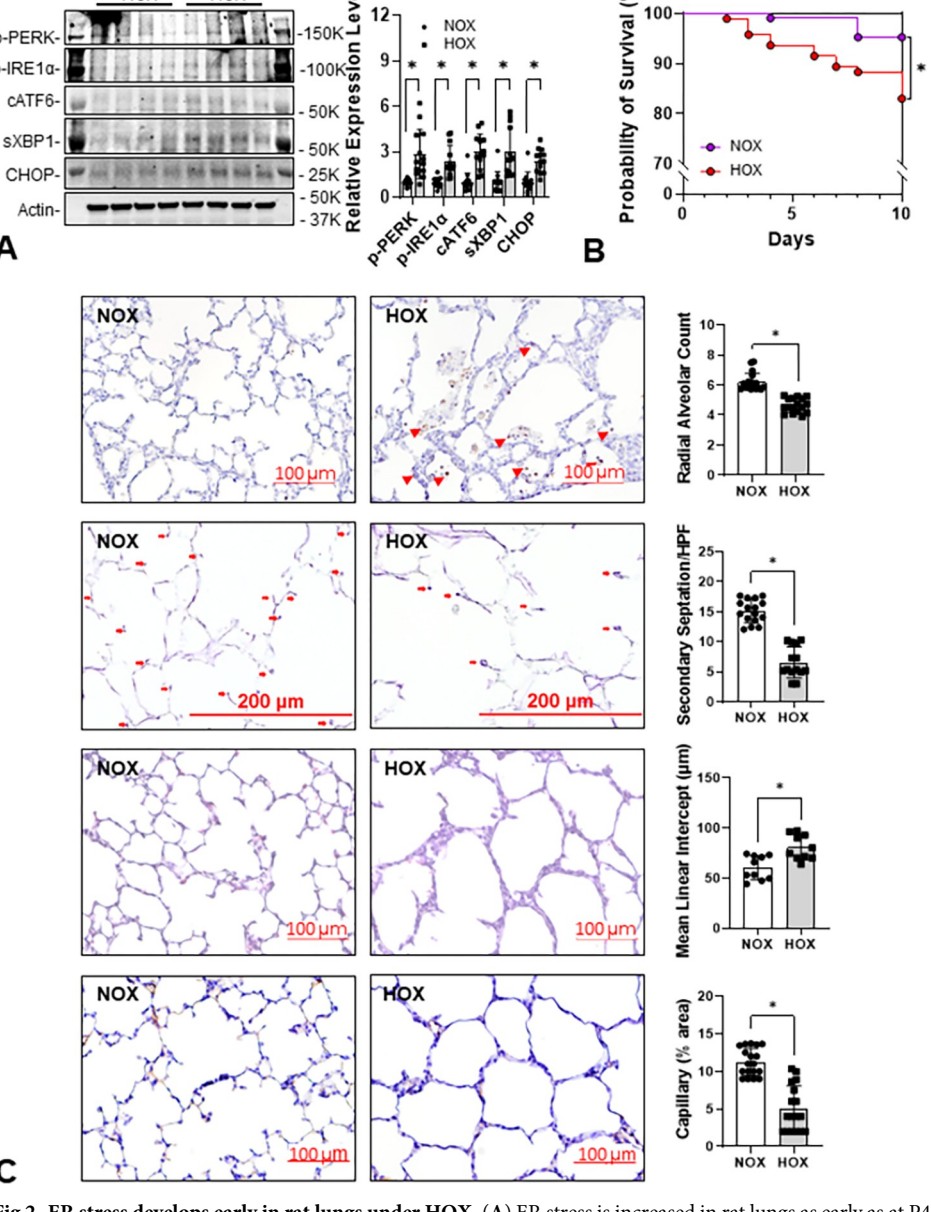

**Fig 2. ER stress develops early in rat lungs under HOX.** (**A**) ER stress is increased in rat lungs as early as at P4 under hyperoxia. The expression levels of five ER stress markers increased by more than 2.4-fold (P-PERK: 2.9±1.6-fold; P-IRE1α: 2.4±1.0-fold; cleaved ATF6: 3.0±1.1-fold; spliced XBP1: 3.0±1.6-fold; CHOP: 2.4±0.8-fold; n = 13 from 3 experiments, 7 males and 6 females in each group, p<0.001 for all markers). (**B**) Survival rate significantly decreases in HOX than NOX rat pups at P10 (84.6% vs 95.6%; p = 0.0047). (**C**) Alveolar simplification, alveolar wall thickening, inflammatory cell infiltration (red arrowhead), decreased secondary septation (red arrow) and capillary formation at P10 are reminiscent of human BPD. *: p<0.05.

## Tun decreases ER-mitochondria interactions in neonatal rat lungs

Mitochondria require continuous support from ER through the mitochondria-associated-membranes (MAM) to maintain function [30]. Hence, Tun may also impair mitochondrial biogenesis and ER-mitochondria interactions [31]. Increased expression of DRP1 and decreased expression of GRP75 and ACSL4, two major MAM proteins [32] are associated with

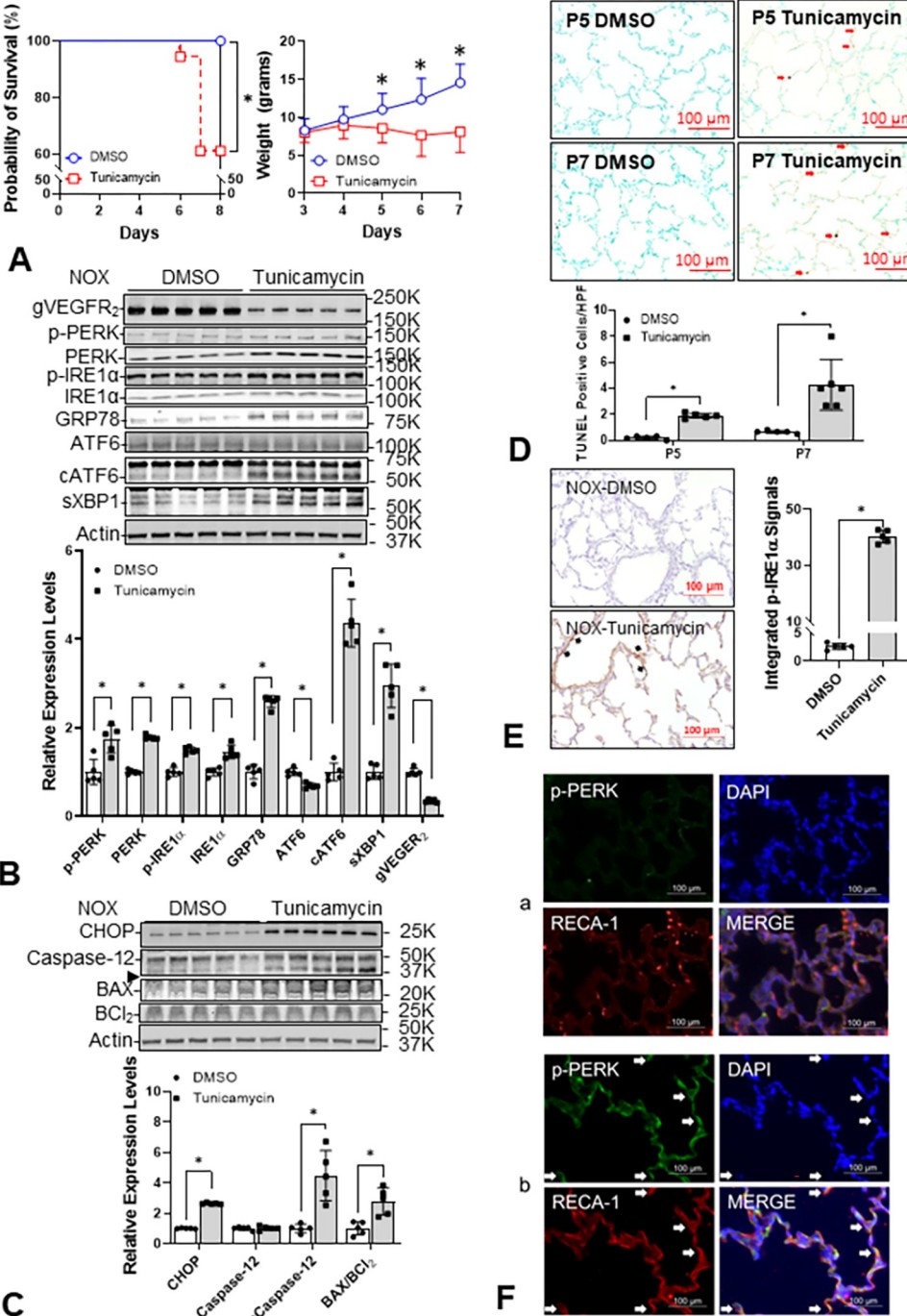

**Fig 3. Tunicamycin (Tun) directly elicits ER stress in NOX neonatal rat lungs.** Rat pups in NOX are treated with one i.p. injection of Tun (0.1 mg/kg), or dimethoxy sulfoxide (DMSO) as control, at P3, and the lungs are obtained at P5 for immunoblots or at P7 for the morphometric study. (**A**) All DMSO-treated pups survived at P7 but 38.9% (7/18) Tun-treated pups died by P7 (p = 0.0077). The body weight difference becomes significant at P5 (p = 0.001586). (**B**) All three ER stress sensors and their active forms (P-PERK: 1.7±0.3-fold, p = 0.004392; PERK: 1.8±0.0-fold, p<0.001; P-IRE1α: 1.5±0.1-fold, p<0.001; IRE1α: 1.5±0.2-fold, p<0.001; cleaved ATF6: 4.4±0.5-fold, p<0.001; n = 5, 2 males and 3 females) and the downstream spliced XBP-1 (2.9±0.5-fold, p<0.001, n = 5) increase by Tun treatment. The endogenous chaperone 78-kDa glucose-regulated protein (GRP78) is increased (2.6±0.1-fold, p<0.001, n = 5) as well. The increased ER stress reasonably explains the decreased expression of N-glycosylated VEGFR2 (0.3±0.0-fold, p<0.001, n = 5). (**C**) The increased levels of CCAAT enhancer-binding protein homologous protein (CHOP, 2.6 ±0.0-fold, p<0.001, n = 5), cleaved caspase-12 (→; 4.5±1.6-fold, p = 0.001708, n = 5), and BAX/BCl₂ ratio (2.8±1.0,

p = 0.003837, n = 5) in Tun treated neonatal rat lungs indicate an increased ER stress-mediated apoptosis. (**D**) The increased in situ TUNEL staining (from 0.4±0.1% to 1.9±0.2% at P5, p<0.001, n = 5; from 0.7±0.1% to 4.22.0% at P7, p = 0.002772, n = 6, 3 per each sex, respectively) in the lungs also supports the increased ER stress-mediated apoptosis by Tun. (**E**) Immunohistochemistry staining shows increased P-IRE1α (40.2±2.3 A.U. *vs.* 2.6±0.5 A.U., p<0.001, n = 5, p<0.05) with the strongest signal located mainly inside of the vasculature (black arrow). (**F**) The increased colocalization (yellow arrow) of P-PERK (green) and rat endothelial cell antigen-1 (RECA-1, red) further indicates the ER stress is increased by Tun (panel b). Blue empty circle and line: DMSO treated control pups; red empty square and line: tunicamycin treated pups.

mitochondrial fission (Fig 4A) [33]. Fragmented mitochondria can be seen in AT2 cells and EC in Tun-treated lungs under electron microscopy (Fig 4B and 4C). These data are similar to what we reported previously in the lungs of HOX-treated neonatal rat pups [13].

## Tun increases MPO+ myeloid cell infiltration into neonatal rat lungs

ER stress induces sterile inflammation via activation of NF-kB or JNK-dependent signaling [14, 34]. Marked increases in MPO protein and MPO+ myeloid cells infiltrating the lungs support the idea that ER stress induced by OS reciprocally aggravates inflammation. Interestingly, a large proportion of MPO protein released into the lung can be seen associated with the EC lining lung blood vessels, suggesting Tun increases neutrophil recruitment and adherence to vascular EC (Fig 4D).

## Tun decreases alveolar formation in neonatal rat lungs

Morphometric analysis of lungs from Tun-treated rat pups revealed that alveolar simplification was increased and radial alveolar counts (RAC) and secondary septation were decreased, and mean linear intercepts (MLI) were increased which is reminiscent of the BPD phenotype. We also saw a marked decrease in staining for vascular ECs, which is consistent with data showing Tun also decreased lung glycosylated VEGFR2 levels (Fig 5).

## TUDCA attenuates Tun-induced ER stress in neonatal rat lungs

The immunoblots demonstrated that TUDCA decreased protein markers for ER stress and ER stress-mediated apoptosis [35] and increased gVEGFR2 in Tun-treated neonatal rat lungs (Fig 6A). The morphometric study showed that TUDCA improved alveolarization and increased endothelial cell density (Fig 6B). Although TUDCA improved overall weight gain in Tun-treated rat pups (n = 20, p<0.05), the daily weights were not significantly different, suggesting that the improved alveolar formation induced by TUDCA may not be the only reason for better nutritional status in the rat pups (Fig 6C).

## TUDCA attenuates HOX-induced ER stress in neonatal rat lungs

After demonstrating that TUDCA attenuated ER stress and alveolar simplification in the lungs from Tun-treated neonatal rat pups, we determined the effect of TUDCA on the lungs from HOX-treated neonatal rat pups. TUDCA treatments decreased ER stress markers and increased gVEGFR2 expression in lungs from HOX-treated neonatal rat pups (Fig 7A). These data indicate TUDCA reduced ER stress and reduced markers of ER stress-mediated apoptosis (Fig 7B). The decrease in IHC staining for P-IRE1α, and the ER caliber of both AT2 cells and EC observed by electron microscopy support the notion that TUDCA attenuated ER stress in the lungs of HOX-treated neonatal rat pups (Fig 7C and 7D).

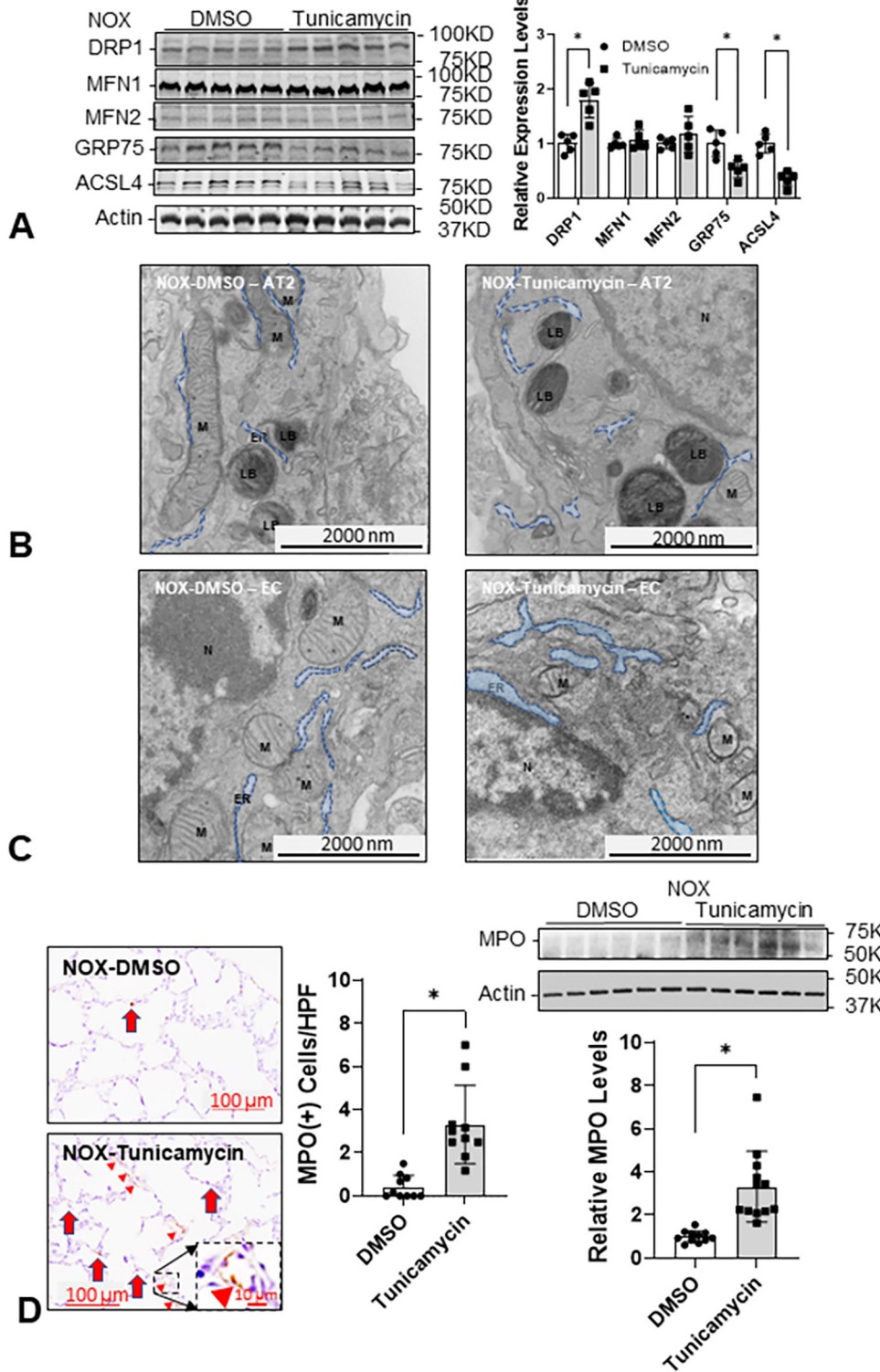

**Fig 4. Tun increases mitochondrial fission, inflammatory cell infiltration, and decreased ER-mitochondria interaction in neonatal rat lungs.** (**A**) There is no significant change in mitochondrial fusion proteins (MFN1 and MFN2) but the expression of fission protein (DRP1) is increased significantly (1.8±0.3-fold, n = 5, 3 males and 2 females per group, p = 0.001152). The decreased expressions of GRP75 (0.5±0.2-fold, p = 0.006767, n = 5) and acyl-CoA synthetase long-chain family member 4 (ASCL4; 0.3±0.1-fold, p<0.001, n = 5) indicate a decreased interaction between ER and mitochondria by Tun. (**B**) The size of mitochondria is smaller, and the distance between ER and

mitochondria is wider in type 2 alveolar cell (AT2) by Tun under electron micrograph. (**C**) Similar structural changes are also seen in the ER and mitochondria of the endothelial cell (EC) under an electron micrograph (20,000x magnification). (**D**) The increased MPO(+) cell infiltration (3.3±4.8 cells/field vs 0.4±0.5 cells/field; n = 11, 5 males and 6 females per group, p<0.001) and MPO protein expression (3.3±0.5-fold, n = 12, 6 for each sex, p<0.001) in Tun treated neonatal lungs. **LB**: the lamellar body which is the distinct subcellular organelle in AT2. **M**: mitochondria. Red arrow: MPO(+) myeloid cells; red arrowhead: MPO staining. *: p<0.05; scale bar = 100 μm for light microscope and = 2,000 nm for electron microscope, respectively.

## TUDCA decreases MPO protein, improves MAM, and alveolar formation in lungs from HOX-treated neonatal rat pups

Based on decreased MPO protein levels in the lungs of TUDCA- and HOX-treated rat pups, TUDCA effectively decreased lung inflammation induced by chronic exposure to HOX. The decrease in DRP1 levels along with an increase in MFN2 expression and an increase in GRP75 and ACSL4 expression suggest fewer mitochondria are undergoing fission, which allows more MAM to form in HOX lungs after TUDCA treatment (Fig 8A). The morphologic studies on lungs from TUDCA-treated HOX rat pups showed that TUDCA markedly improved alveolar formation and angiogenesis (Fig 8B).

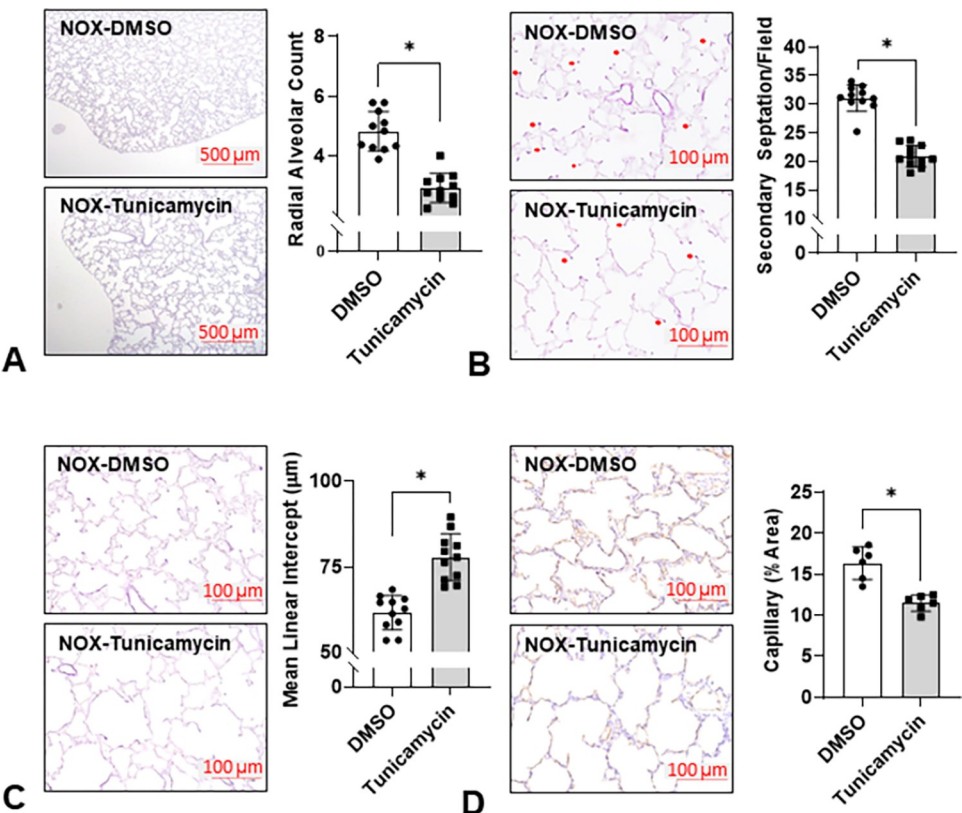

**Fig 5. Tun impairs alveolarization in NOX neonatal rat lungs.** Rat pups in room air receiving a single injection of either Tun or DMSO were euthanized at P7 for lung histology. The decreased radial alveolar counts (RAC; 2.9±0.5 *vs* 4.8±0.7, n = 11, 5 males and 5 females each group, p<0.001) (**A**), increased mean linear intercepts (MLI; 77.8±6.8 μm *v.s.* 61.6±5.0 μm, n = 11, p<0.001) (**B**), decreased number of secondary septation (21.0±1.8 *vs* 31.1±2.4, n = 11, p<0.001) (**C**), and decreased capillary area (14.5±1.0% *vs* 16.4±2.0%, n = 6, 3 for each sex, p<0.001) (**D**) all indicate an impaired alveolarization. *: p<0.05.

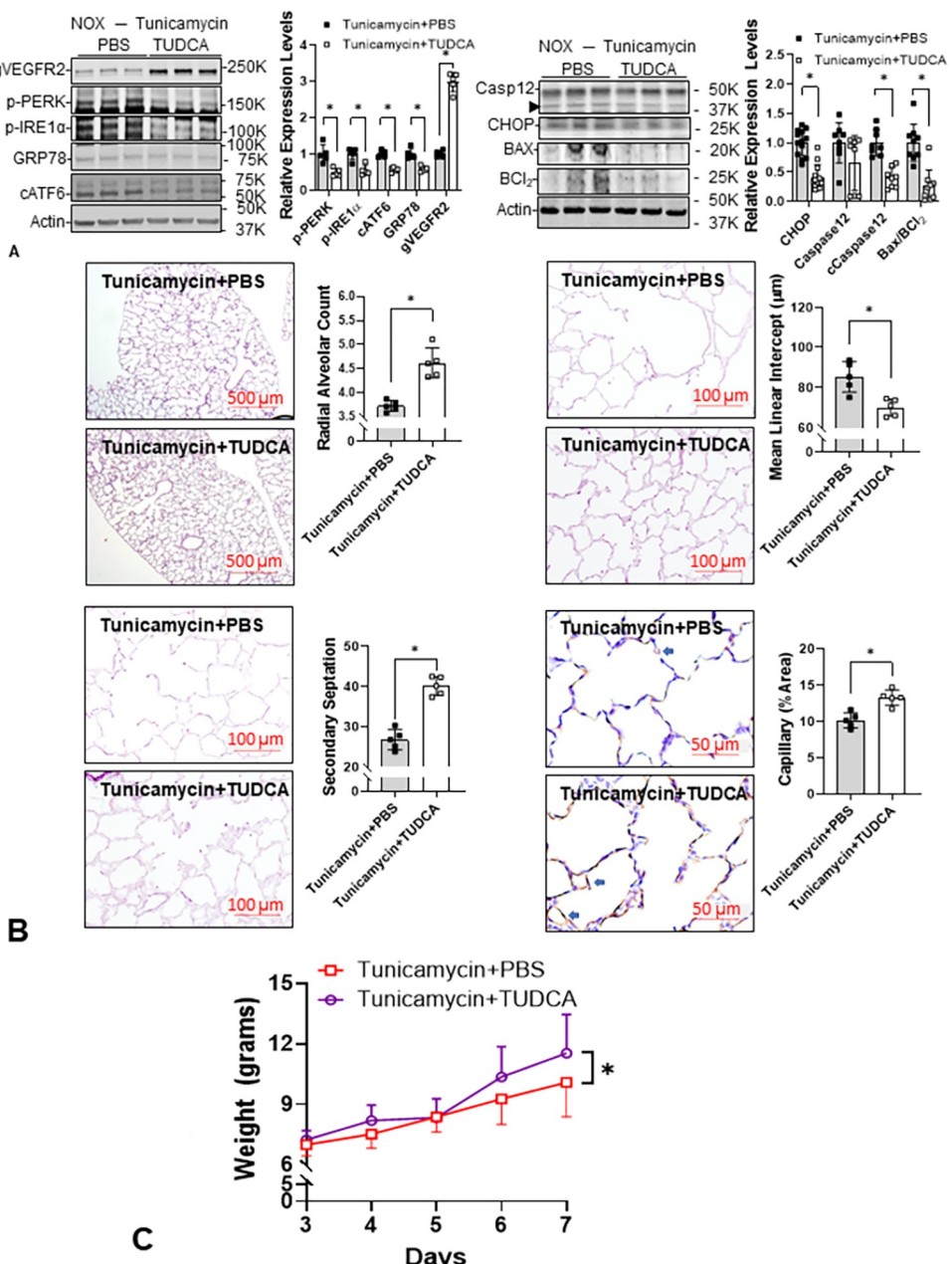

**Fig 6. Tauroursodeoxycholic acid (TUDCA) attenuates ER stress and improves alveolarization in Tun-treated neonatal rat lungs.** (**A**) The expression levels of ER stress sensors (P-PERK: 0.5±0.1-fold, p = 0.002814; P-IRE1α: 0.5 ±0.1-fold, p = 0.001463; cleaved ATF6: 0.6±0.1-fold, p<0.001, n = 9, 4 males and 5 females), GRP78 (0.6±0.1-fold, p<0.001, n = 9), CHOP (0.4±0.2-fold, p<0.001, n = 9), cleaved caspase-12 (0.5±0.1-fold, n = 13, 6 males and 7 females), and BAX/BCl$_2$ ratio (0.3±0.3-fold, p<0.001, n = 12, 6 for each sex) are all decreased while the glycosylated VEGFR2 is increased (3.0±0.3-fold, p<0.001, n = 9) by TUDCA in Tun treated neonatal lungs (n = 9–13) at P5. (**B**) Lung morphometric parameters including radial alveolar count (4.6±0.3 *vs* 3.7±0.1, p<0.001, n = 5, 2 males and 3 females), secondary septation (40.1±2.3 *vs* 26.8±2.5, p<0.001, n = 5), mean linear intercept (70.1±4.2 μm *vs* 85.4 ±7.6 μm, p = 0.0043, n = 5), and endothelial cell staining (11.3±1.1% *vs* 10.2±1.0%, p = 0.0015, n = 5) are all improved by TUDCA in Tun-treated neonatal rat lungs. (**C**) The improved alveolar formation cannot be explained completely by nutritional status since body weight does not improve in the TUDCA-treated rat pups until P7. *: p<0.05.

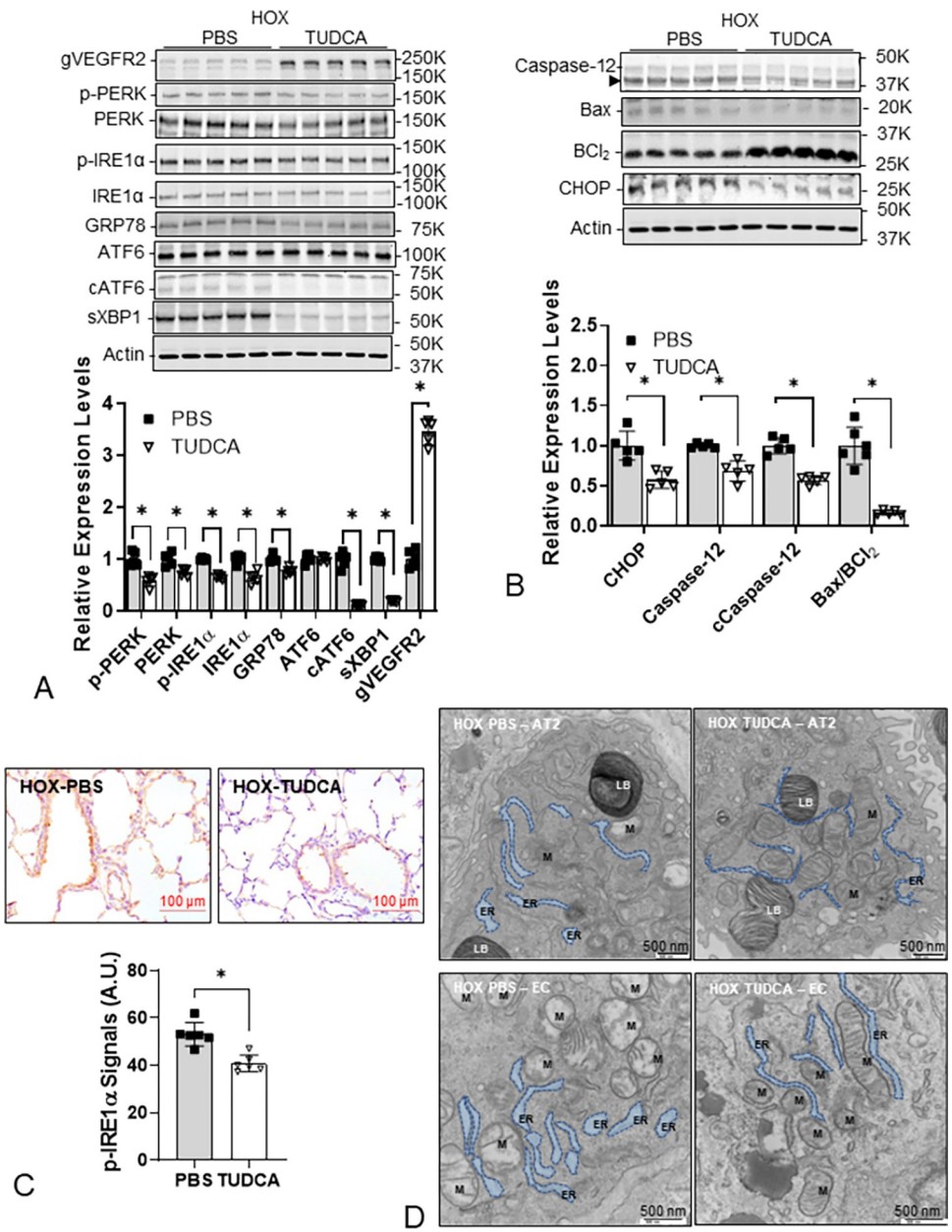

**Fig 7. TUDCA decreases ER stress in HOX neonatal rat lungs.** (**A**) The expression levels of ER stress markers (P-PERK 0.6±0.1-fold, p<0.001, n = 5; PERK 0.7±0.1-fold, p = 0.00364, n = 5; P-IRE1α 0.6±0.0-fold, p<0.001, n = 5; IRE1α 0.6±0.1-fold, p<0.001, n = 5; GRP78 0.8±0.1-fold, p<0.001, n = 5; cleaved ATF6 0.1±0.0-fold, p<0.001, n = 5; spliced XBP1 0.3±0.0-fold, p<0.001, n = 5; 2 males and 3 females) are all decreased while the N-glycosylated VEGFR2 is increased (3.5±0.2-fold, p<0.001, n = 5) in HOX neonatal lungs at P10 indicating TUDCA can attenuate hyperoxia-induced ER stress. (**B**) TUDCA also decreases ER stress-mediated apoptosis as evidenced by the decreased CHOP (0.6 ±0.1-fold, p = 0.002001, n = 5), caspase-12 (0.7±0.1-fold, p<0.001, n = 5), cleaved caspase-12 (0.6±0.1-fold, p<0.001, n = 5), and BAX/BCl$_2$ ratio (0.2±0.0-fold, p<0.001, n = 5). (**C**) In IHC stain, P-IRE1α levels are decreased (40.8±3.5 A. U. *vs* 53.1±5.0 A.U., p<0.001, n = 6, 3 for each sex) in chronic hyperoxia exposed neonatal rat lungs by TUDCA. (**D**) The dilated ER structure, ER-mitochondria separation, and mitochondrial fission are seen both in AT2 (20,000x magnification) and EC (30,000x magnification) while TUDCA partially reverses the structural changes under the electron microscope. **LB**: lamellar body in AT2. **M**: mitochondria. Scale bar = 100 μm for light microscope and = 500 nm for electron microscope. *: p<0.05.

## TUDCA improves bodyweight gain and alveolarization in lungs from HOX neonatal rats

To confirm the safety of TUDCA, we first mixed four litters of rat pups and then randomly assigned them per sex into NOX or HOX, with or without TUDCA treatments. The nursing

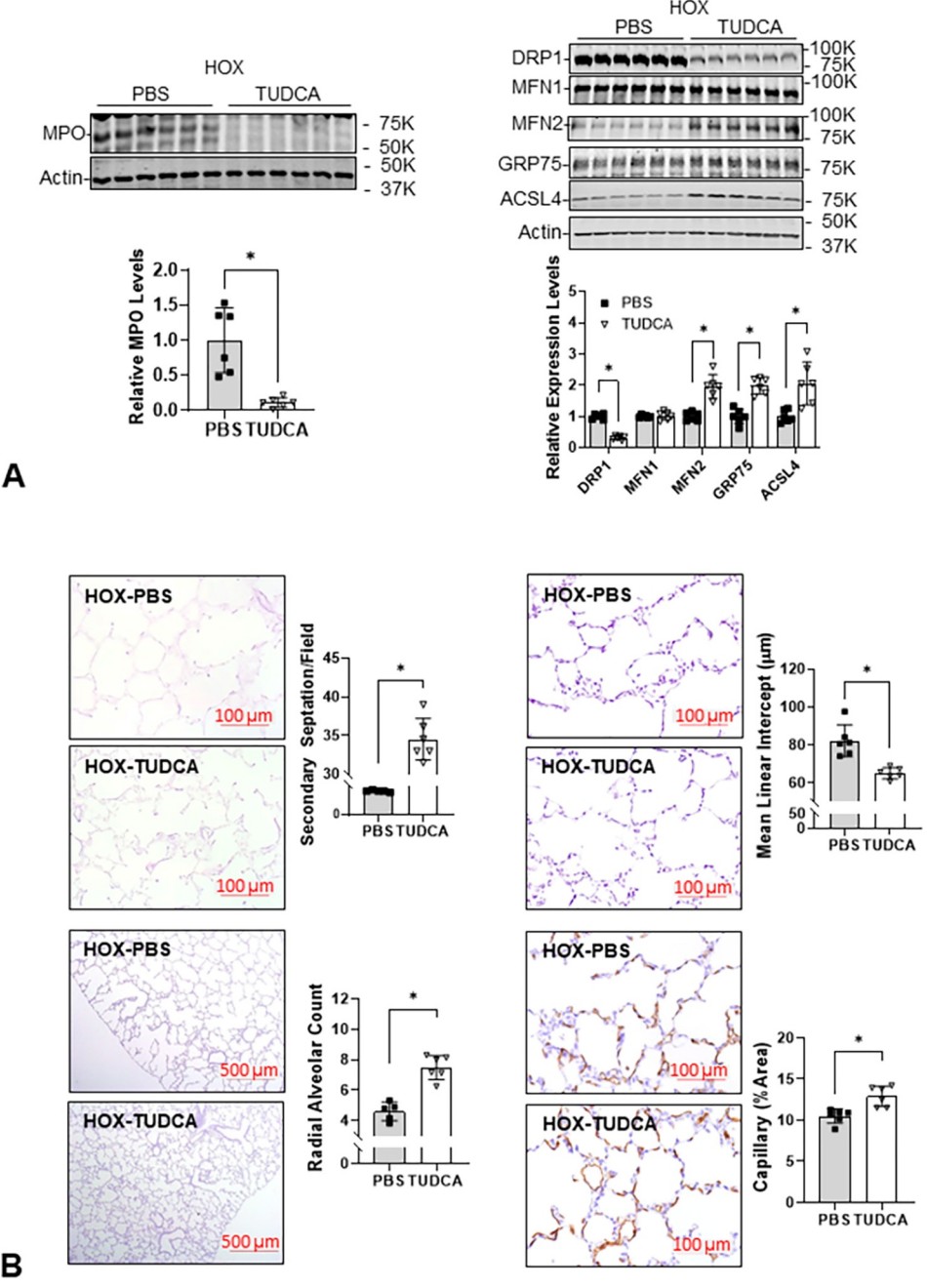

**Fig 8. TUDCA decreases MPO with improved mitochondrial biogenesis, mitochondria-ER contact, and alveolar formation of HOX neonatal rat lungs.** (**A**) The increased MPO levels in HOX neonatal lungs are decreased by TUDCA treatment (0.1±0.1-fold; n = 6, p<0.05). The decreased DRP1 levels (0.3±0.1-fold) with increased MFN2 levels (2.0±0.4-fold), GRP75 levels (2.0±0.3-fold), and ASCL4 levels (2.1±0.7-fold) indicate an improved ER-mitochondrial interaction (n = 6, p<0.05). The increased RAC (7.3±0.8 vs. 4.6±0.6) and secondary septation (34.5±3.0 vs. 24.8±1.1) with decreased MLI (65.1±2.9 μm vs. 82.3±8.4 μm) and increased capillary area (12.9±1.2% vs. 10.5±0.9%) demonstrate a protective effect of TUDCA to the HOX neonatal lungs (n = 6, p<0.05, 3 for each sex). *: p<0.05.

dams were alternated daily between NOX and HOX. The survival rate was significantly lower in the HOX-PBS group than in the other three study groups. The weight gain began to decrease at P5 in both HOX groups, but by P8 the weight gain in HOX groups started to catch up, although it was still lower than in the two NOX groups. There was no difference in survival rate and weight gain in the NOX group treated with daily TUDCA, indicating that TUDCA was well-tolerated without toxicity at the daily dosage we used (Fig 9A). Lung morphometry revealed that TUDCA had little effect on alveolar formation in NOX rat pups but did improve alveolar formation in the lungs of HOX-treated rat pups (Fig 9B).

## KYC decreases ER stress in the lungs of HOX-treated neonatal rats and improves alveolarization in the lungs of Tun-treated rats

We recently reported that KYC improves alveolar formation in the lungs of HOX-treated rat pups by repurposing MPO from a toxic peroxidase into a quasi-catalase which degrades hydrogen peroxide to decrease oxidative stress [10]. The decreased markers for ER stress and ER stress-mediated apoptosis in the lungs of HOX neonatal rat pups treated with KYC are consistent with the idea that KYC attenuates ER stress by repurposing MPO (Fig 10A). The decrease in ER caliber and increase in contact between the ER and mitochondria in EC and AT2 cells in lungs from KYC-treated HOX neonatal rat lungs further supports our interpretation (Fig 10B). We randomized Tun-treated pups to receive daily KYC or PBS injections to confirm our assumption. Again, KYC also decreased ER stress and ER stress-mediated apoptosis markers with increased gVEGFR2 in lungs isolated from Tun-treated neonatal rats (Fig 10C). Increased GRP75 and ACSL4 suggested better mitochondria-ER contact (Fig 10D). The decreases in DRP1 and MPO with KYC in the lungs from Tun-treated neonatal rat pups are similar to the effects of TUDCA (Fig 10E). Although KYC failed to improve the weight, it still improved alveolar formation in the Tun-treated rat pups (Fig 10F).

## Discussion

### ER stress is increased in BPD

It is well accepted that prematurely born infants rapidly manifest OS after birth due to their underdeveloped antioxidant defense system and the rapid surge of oxygen tension provided during medical treatment [36]. Several adaptive survival mechanisms are activated to cope with the abrupt increase in OS, including the UPR (a response to shifts in chaperone activity), which results in increased ER stress and a reduction in protein synthesis to restore proteostasis [37]. ER stress and OS are two interrelated states involved in many lung diseases [38]. It has recently been reported that increased ER stress is seen in the lungs of several established BPD animal models [11–13, 39]. Using the lung explant model of GRP78 knockout fetal mice, Flodby et al. demonstrated that the increased ER stress and impaired alveolar formation could be reversed by TUDCA, indicating a potential therapeutic strategy for preventing BPD development [40]. However, how OS is related to ER stress and how OS and ER stress combine to induce BPD remains unclear.

This study first demonstrated an increased P-IRE1α IF of AT1 cells in human BPD lungs compared to age- and sex-matched controls, indicating an increased ER stress in human BPD lungs (Fig 1A). As AT1 is the primary cell type for gas exchange, this change may contribute to the increased oxygen requirement in BPD infants. To investigate the role of ER stress in the development of BPD, we treated neonatal rat pups with either Tun or HOX. P-IRE1α IF and transcriptomic data from studies here, and the proteomic data from our previous report [13], all supported the idea that HOX increased ER stress in the lungs of rat pups at P10 (Figs 1B, 1C and S1). We

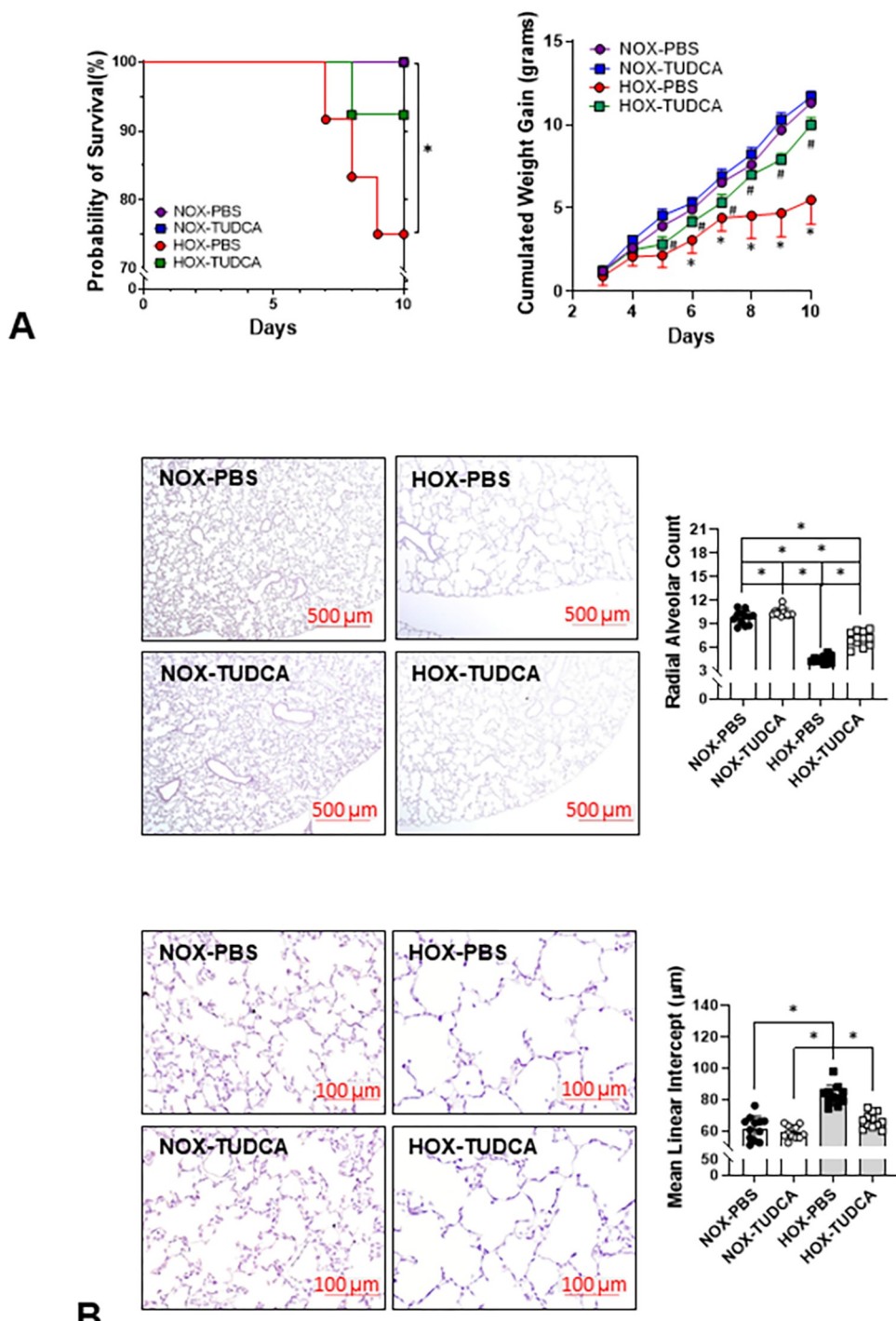

**Fig 9. TUDCA improves weight gain and alveolar formation in HOX rat pups.** (**A**) All NOX pups survived (26/26, 14 males and 12 females), while three HOX pups (3/25, 13 males and 12 females) died before P10. The survival rate has a trend of improvement by TUDCA (Log-rank test for trend, p = 0.02). The weight gains start to decrease from P5 in HOX pups, while TUDCA improves the weight gains after P7. (**B**) The decreased RAC in the HOX neonatal lungs (4.4 ±0.5 *vs.* 9.8±0.9; n = 10 (5 for each sex) and 12 (6 for each sex, respectively) was partially reversed by TUDCA (7.1±1.6, n = 11, 5 males and 6 females). Interestingly, TUDCA treatment does lead to a small but significantly increased RAC in normoxic neonatal lungs (10.4±0.5; n = 12, 6 for each sex). The increased MLI in HOX neonatal lung (82.6±6.5 μm *vs.* 61.6±7.5 μm, n = 10 and 12, respectively) was also partially reversed by TUDCA (66.5±4.8 μm, n = 12). TUDCA does not significantly affect MLI in NOX neonatal lungs (59.2±4.1 μm, n = 12). Purple circle and line: NOX PBS; blue square and line: NOX TUDCA; red circle and line: HOX PBS; green square and line: HOX TUDCA. *: p<0.05 between groups; #: p<0.05 in HOX between pups treated with TUDCA and PBS.

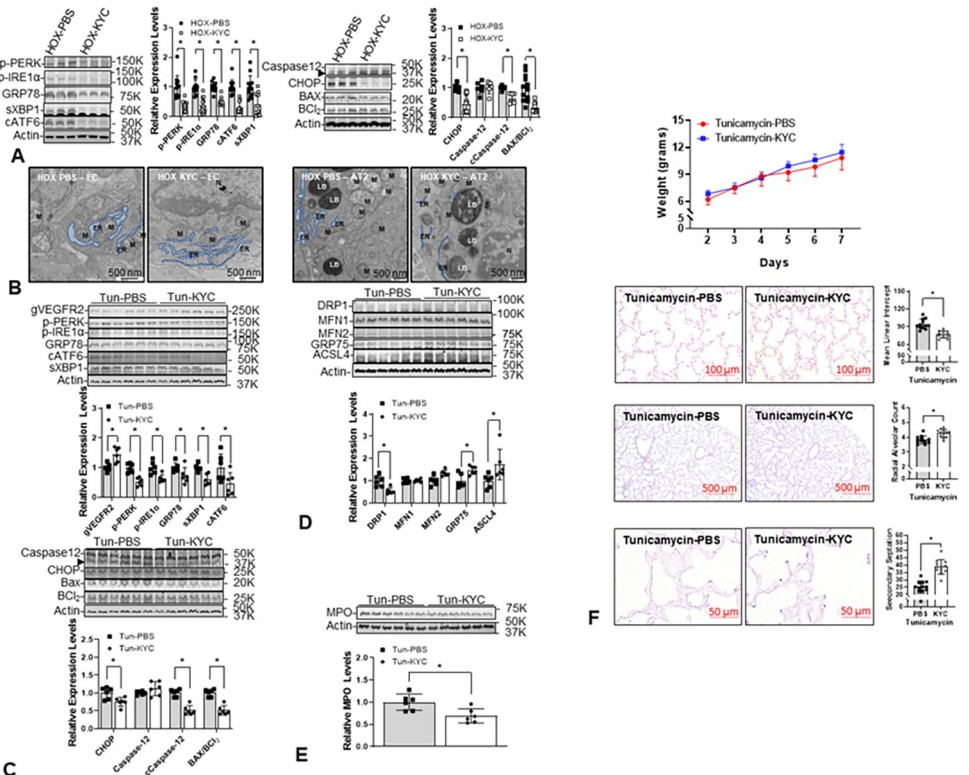

**Fig 10. N-acetyl-lysyltyrosylcysteine-amide (KYC) attenuates ER stress in HOX and Tun-treated neonatal lungs with improved alveolar formation.** (**A**) All ER stress markers in the HOX lungs (P-PERK 0.4±0.2-fold; P-IRE1α 0.4 ±0.2-fold; GRP78 0.5±0.1-fold; cleaved ATF6 0.3±0.2-fold; spliced XBP1 0.4±0.4-fold; CHOP 0.4±0.3-fold) decrease significantly by KYC (n = 12, p<0.05, 6 for each sex). (**B**) KYC improves the ER dilation in EC and AT2 of the HOX lungs. (**C**) KYC decreases ER stress markers (P-PERK 0.5±0.1-fold; P-IRE1α 0.6±0.1-fold; GRP78 0.7±0.3-fold; spliced XBP1 0.6±0.2-fold; cleaved ATF6 0.5±0.4-fold) and ER stress-mediated apoptosis (CHOP 0.8±0.1-fold; cleaved caspase-12 0.5±0.1-fold; BAX/Bcl2 ratio 0.7±0.1-fold) in Tun-treated neonatal lungs (n = 6, p<0.05). But levels of N-glycosylated VEGFR2 are significantly increased (1.4±0.3-fold) by KYC. (**D**) KYC decreases the DRP1 expressions (0.5 ±0.1-fold) and improves the ER-mitochondria contact, as shown by the increased expressions of GRP75 (1.5±0.2-fold) and ACSL4 (1.7±0.7-fold). (**E**) KYC significantly decreases MPO expression levels (0.7±0.1-fold) in the Tun-treated lungs. (**F**) Although KYC does not improve the weight gain in Tun-treated rat pups, it still improves the alveolar formation (MLI 76.8±5.9 vs. 94.2±8.4 μm; RAC 4.3±0.3 vs. 3.8±0.3; secondary septation 38.7±4.9 vs. 25.2±4.1; n = 12, p<0.05). **LB**: lamellar body; **M**: mitochondria. *: p<0.05; scale bar = 500 nm.

even observed increased expression of ER stress markers and critical proteins in the antioxidant defense system at P4 (Figs 2A and S3). Persistent activation of antioxidant enzymes was also seen at P10 (S2 Fig and S2 Table). This increase in ER stress is essential because P4 is before alveolo-genesis and confirms findings by others showing that early responses are due to increases in anti-oxidant defense gene expression. In contrast, later responses decrease antioxidant defense gene expression [10]. In this context, it is important to note that KYC not only repurposes MPO into a quasi-catalase, but it also derepresses Keap-1 allowing the release and activation of Nrf2 to increase the expression of antioxidant defense enzymes [10]. As the onset and development of BPD are temporal additional studies are required to determine the time-dependent effects of HOX on MPO, OS, and ER stress in neonatal rat lungs.

ER is the subcellular organelle that provides a unique milieu for synthesizing proteins, lip-ids, and cholesterols, metabolizing carbohydrates, and storing calcium [41]. The ER plays a critical role in alveologenesis by (1) maintaining mitochondrial function; (2) correcting post-translational modification of the growth factors and their receptors; (3) limiting the generation

of oxidative stress; and (4) limiting inflammation [42]. In this context, it is important to note that mitochondrial dysfunction [43], impaired angiogenesis [44], and inflammation [2] have all been identified as major contributors to BPD. ER provides calcium and biomolecules through MAM to support mitochondrial function [45]. Previously we have reported the increased DRP1 expression in HOX rat BPD lungs compared to NOX rat lungs [13]. In this study, we demonstrated that DRP1 expression increased and the expression of GRP75 and ACSL4 decreased in Tun-treated rat lungs, which reasonably explains the fragmented mitochondria and reduced contact between the ER and the mitochondria observed under the electron microscope. Likewise, a decrease in gVEGFR2 in neonatal rat pups treated with HOX or Tun explains why lung capillary density count was reduced (Fig 4A–4C).

There are at least four different mechanisms by which ER stress induces and accelerates BPD. First, chronic ER stress induces apoptosis; apoptotic cells release HMGB1, which increases myeloid cell recruitment and vascular inflammation. Second, the myeloid cell recruited to the lung in response to this HMGB1 is an added cellular source of MPO, generating toxic oxidants that induce cell necrosis. Third, the reactive oxidants generated during efforts to refold proteins under ER stress in an attempt to restore proteostasis represent another cellular source of OS. Fourth, CHOP and caspase-12, by mechanisms unrelated to the others, can activate anti-angiogenic and likely anti-alveologenic signaling pathways to impair lung development [46]. Such diverse and extensive biological effects suggest that ER stress adds a second LOOP to the cycle of destruction we described earlier [10].

GRP78 is an endogenous chaperone that facilitates protein folding, assembly, and transports synthesized proteins across the ER [47], which is considered the master regulator of ER stress through stabilizing the PERK and IRE1α [48]. Excessive accumulation of unfolded proteins in ER will cause a shift in GRP78 chaperone activity, separating GRP78 from PERK and IRE1α. Accordingly, PERK and IRE1α will dimerize and auto-phosphorylate to activate the UPR to mitigate ER stress. A global GRP78 knockout is lethal [49], while epithelial-specific GRP78 knockout mice have impaired *in utero* lung formation with less than an 8% chance to survive to birth [40]. These facts indicate that UPR is a survival mechanism with increased GRP78 expression to resume the ER function under stress. Assisting the chaperone activity should help stressed cells resume homeostasis. Our TUDCA data support this idea since TUDCA attenuated the alveolar simplification in HOX neonatal rats and Tun-treated neonatal rats (Figs 6, 8 and 9). Although the poor weight gain may be a plausible explanation for the impaired alveolar formation in Tun-treated neonatal rats, the improved alveolar formation after TUDCA treatment with little weight improvement strongly suggests that ER stress is the primary cause of the Tun-induced alveolar simplification.

## Interactions between OS and ER stress play a mechanistic role in BPD

From our previous work and the data presented here, we think HOX induces neonatal lung damage in the following way. Initially, the supplement oxygen may only cause lung endothelial cells to generate superoxide anion [50] and resident myeloid cells to generate toxic oxidants [51]. Although the levels and types of toxic oxidants generated during onset by resident myeloid cells may be low, chronically increased levels can and do result in cell injury and death [52]. When lung cells die, however, they passively release HMGB1, a danger-associated molecular pattern (DAMP) molecule that has potent cytokine- and chemokine-like properties [53]. Although HMGB1 from dead and dying lung cells may be released in low quantities at the onset, it is well known that the amount of extracellular HMGB1 released after a severe injury is directly proportional to the number of dead and dying lung cells [54]. Accordingly, as BPD

progresses, neonatal lungs will experience even greater cell injury and cell death, resulting in more HMGB1 release, especially when the number of MPO⁺ myeloid cells recruited to the injured lung increase [10].

The ER is a subcellular organelle where secretory and membrane proteins are synthesized, folded, and modified. ER homeostasis is maintained by an endogenous chaperone 78-kD glucose-regulated protein (GRP78) which stabilizes protein kinase RNA (PKR)-like ER kinase (PERK) and inositol-requiring enzyme 1α (IRE1α) to prevent their homo-dimerization and autophosphorylation [37]. These latter events are the initial steps in the unfolded protein response (UPR) when GRP78 is recruited to stabilize unfolded proteins accumulating in the ER [55]. Under excessive OS, cells use UPR to maintain or restore proteostasis, unfortunately, when efforts fail to restore proteostasis, marked increases in non-specific inflammatory responses are observed [11, 42]. One of the by-products of the UPR is the generation of reactive oxygen species that adds to cellular OS [56, 57]. When OS is prolonged, or unopposed, the activated UPR will induce apoptosis via activating Jun kinase (JNK) and CCAAT/enhancer-binding proteins homologous protein (CHOP) [58, 59].

OS can negatively impact angiogenesis. Sprouting angiogenesis is essential for forming alveoli and lung development and repairing injured lungs [60]. Successful angiogenesis requires endothelial cells to express vascular endothelial cell growth factor receptor type 2 (VEGFR2) in response to the VEGF released by the neighboring cells [61] and adequate mitochondrial function. Post-translational VEGFR2 modification [62] and assisting mitochondrial function [45, 63] are two mechanisms that ER is needed for angiogenesis. In summary, ER stress can impair angiogenesis by increasing endothelial cell death [64, 65], decreasing VEGFR2 N-glycosylation [29], and diminishing mitochondria oxidative phosphorylation [30]. As angiogenesis plays such a critical role in forming alveoli in neonatal lungs [44], any increase in ER stress should limit alveolarization. The non-specific inflammation associated with ER stress can also be expected to contribute the BPD.

## Chemical chaperone as potential treatment for BPD

TUDCA is a taurine-conjugated derivative of ursodeoxycholic acid (UDCA) that is more hydrophilic and less toxic than UDCA [66]. For decades, UDCA has been used as an off-label treatment for neonatal cholestasis [67]. TUDCA was first recognized as a chemical chaperone when it was introduced to treat cholestatic liver disorders [15], type 2 diabetes [68], and retinal degeneration [69]. A clinical study showed that it is safe to use TUDCA to prevent cholestasis in premature infants [19]. TUDCA has also been shown to improve alveolar formation in the *ex vivo* fetal lung explant cultures of epithelial-specific GRP78 knockout mice [40]. With this information as background, it is not surprising that TUDCA can markedly attenuate ER stress and improve alveolar formation in the lungs of Tun-treated neonatal rat pups.

Bolstered by the promising effects of TUDCA on the lungs of Tun-treated rat pups, we treated HOX neonatal rat pups with TUDCA to determine if it could attenuate the severity of BPD in HOX neonatal rats without inducing toxicity. In the HOX BPD rat model, we showed TUDCA effectively decreased ER stress and MPO levels, and improved the survival rate, alveolar formation, and weight gain. Importantly, no adverse effect was identified resulting from TUDCA treatments of NOX neonatal rats. This finding is encouraging and suggests that TUDCA is a promising therapeutic for preventing BPD and should be further investigated. We did not study whether TUDCA provides similar protection in combined HOX-Tun treated rat pups believing the data already demonstrated the contributing role of ER stress in BPD (Figs 7–9).

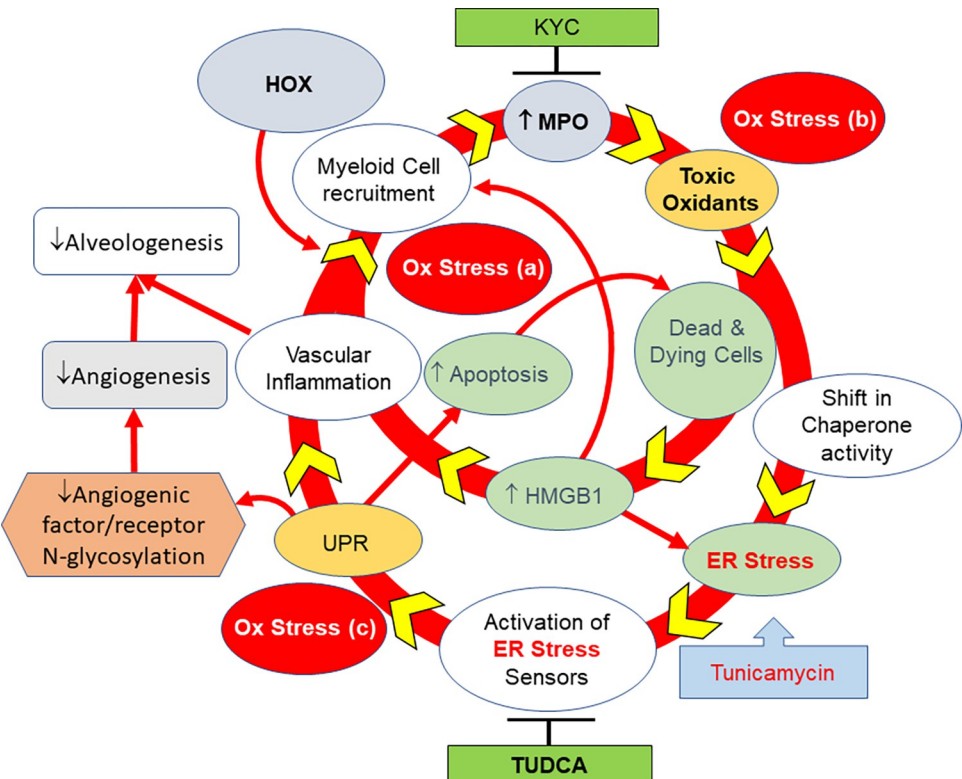

**Fig 11. ER stress plays a critical role in propagating the cycle of destruction to cause BPD.** Supplemental oxygen initiates the cycle of destruction by generating oxidative stress and recruiting inflammatory cells. MPO released by the infiltrated myeloid cell generates toxic oxidants that produce ER stress. ER stress propagates the destruction cycle by causing more oxidative stress and inflammation. ER stress impairs angiogenesis by decreasing the N-glycosylation of angiogenic factors and their corresponding receptors, such as VEGFR2. Tunicamycin inhibits alveolar formation by inducing ER stress. TUDCA, a chemical chaperone, attenuates alveolar simplification by reducing ER stress elicited by either supplemental oxygen or tunicamycin. Our findings strongly indicate a mechanistic role of ER stress in BPD. The protective effect of KYC is believed to be mediated by its activity in repurposing MPO into a quasi-catalase that decreases the generation of toxic oxidants by MPO.

Since BPD is a complex disease that includes multiple mechanisms of OS and inflammation, the most effective strategy for preventing BPD is to use multiple agents to inhibit multiple targets or a systems pharmacology agent that inhibits various targets. To our knowledge, KYC is the first systems pharmacology agent that effectively inhibits multiple targets in the innate immune system. We have recently reported that KYC repurposes MPO to enhance hydrogen peroxide catabolism and exploits MPO peroxidase activity to be oxidized into a novel anti-inflammatory KYC thiyl radical that inactivates HMGB1 and derepresses Keap-1 to activate Nrf2 to enhance antioxidant gene expression [10]. These unique pharmaceutical properties suggest KYC is a first-in-class systems pharmaceutical agent for preventing lung injury resulting from a dysregulated innate immune system [10]. Studies here show that KYC attenuates ER stress in HOX and Tun-treated neonatal lungs and improves alveolar formation in Tun-treated neonatal lungs, with decreased MPO levels, even without improving body weight (Fig 10). As no evidence indicates KYC is a chemical chaperone, the most likely explanation for KYC inhibiting BPD is that dysregulated innate immunity, involving MPO, is upstream of ER stress and restoring physiological balance to the innate immune system breaks the cycle of destruction.

*In conclusion*, ER stress plays a causal role in developing BPD. It evolves from the chronic states of OS and inflammation generated by the cycle of destruction (Fig 11). After ER stress develops, it increases the severity and accelerates BPD development primarily in neonates who lack sufficient antioxidant defenses. ER stress expands the cycle of destruction in HOX neonatal lungs that we previously proposed [9]. TUDCA appears to have therapeutic value for protecting neonatal lungs against BPD. As KYC decreases ER stress and increases antioxidant defenses in the lungs of HOX neonatal rat pups suggests BPD is caused by a cycle of destruction that contains more than one loop. Future studies will investigate the effects of TUDCA and KYC on BPD.

## Supporting information

**S1 Fig. Designs for animal experiments.** Rat pups from 2–4 dams were mixed and then randomly allocated into different treatment groups at P1. After randomization, the pups were raised in either NOX or HOX with nursing dams. Tunicamycin 0.01 mg/kg was given i.p. once at P3. Tauroursodeoxycholic acid 100 mg/kg/dm i.p., or KYC 10 mg/kg/d i.p., was given once daily starting at P2. Lungs were obtained at P7 for tunicamycin studies but at P10 for HOX studies.
(TIF)

**S2 Fig. ER stress-related biological processes are enriched in HOX neonatal rat lungs.** Multiple gene enrichment analysis shows multiple ER stress-related biological processes are annotated by ToppCluster according to the Gene Ontology Biological Processes (GO-BP).
(TIF)

**S3 Fig. NRF2-related pathways are enriched in HOX neonatal rat lungs.** Multiple gene enrichment analysis shows ToppCluster annotates two NRF2-related pathway processes according to the WikiPathways.
(TIF)

**S4 Fig. Downstream NRF2-related antioxidants are upregulated in HOX neonatal rat lungs.** Expressions of NRF2 downstream antioxidants (thioredoxin-1, glutathione-S-transferase, and heme oxygenase-1) are increased in HOX neonatal rat lungs as early as P4; some of them (GST1 and HO1) persist until P10, or (HO1) even persist after recovery in room air for 11 days at P21.
(TIF)

**S1 Table. Age of death, sex, and primary diagnoses of the autopsied subjects.** Detailed information was not available for this age- and sex-matched deidentified specimens. There were 10 cases, 6 males and 4 females, for each group. Half (5 out of 10) of the matched controls died of inoperable complex congenital heart diseases, and none received extensive mechanical ventilator or supplemental oxygen support.
(PDF)

**S2 Table. Transcriptomes identified by ToppCluster to be enriched in NRF2-ARE regulation and NRF2 pathways.**
(PDF)

**S1 Raw images.**
(PDF)

## Author Contributions

**Conceptualization:** Kirkwood A. Pritchard, Jr., Adeleye J. Afolayan, Jason Jarzembowski, Billy W. Day, Stephen Naylor, G. Ganesh Konduri, Ru-Jeng Teng.

**Data curation:** Xigang Jing, Michelle Teng, Clive Wells, Shuang Jia, Jason Jarzembowski, Ru-Jeng Teng.

**Formal analysis:** Kirkwood A. Pritchard, Jr., Xigang Jing, Stephen Naylor, Ru-Jeng Teng.

**Funding acquisition:** Kirkwood A. Pritchard, Jr., Martin J. Hessner, G. Ganesh Konduri, Ru-Jeng Teng.

**Investigation:** Xigang Jing, Michelle Teng, Clive Wells, Ru-Jeng Teng.

**Methodology:** Kirkwood A. Pritchard, Jr., Xigang Jing, Michelle Teng, Shuang Jia, Martin J. Hessner, Ru-Jeng Teng.

**Project administration:** Kirkwood A. Pritchard, Jr., Billy W. Day, Stephen Naylor, Ru-Jeng Teng.

**Resources:** Kirkwood A. Pritchard, Jr., Adeleye J. Afolayan, Jason Jarzembowski, G. Ganesh Konduri, Ru-Jeng Teng.

**Software:** Shuang Jia, Ru-Jeng Teng.

**Supervision:** Kirkwood A. Pritchard, Jr., Billy W. Day, Stephen Naylor, Ru-Jeng Teng.

**Validation:** Kirkwood A. Pritchard, Jr., Clive Wells, Shuang Jia, Ru-Jeng Teng.

**Visualization:** Kirkwood A. Pritchard, Jr., Clive Wells, Shuang Jia, Stephen Naylor.

**Writing – original draft:** Kirkwood A. Pritchard, Jr., Xigang Jing, Stephen Naylor, Ru-Jeng Teng.

**Writing – review & editing:** Kirkwood A. Pritchard, Jr., Adeleye J. Afolayan, Jason Jarzembowski, Billy W. Day, Stephen Naylor, Martin J. Hessner, G. Ganesh Konduri, Ru-Jeng Teng.

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
