## [Decision Letter · Decision Letter 0]

20 Apr 2022

PONE-D-22-06017Role of endoplasmic reticulum stress in impaired neonatal lung growth and bronchopulmonary dysplasiaPLOS ONE

Dear Dr. Teng,

Thank you for submitting your manuscript to PLOS ONE. After careful consideration, we feel that it has merit but does not fully meet PLOS ONE’s publication criteria as it currently stands. Therefore, we invite you to submit a revised version of the manuscript that addresses the points raised during the review process. Please submit your revised manuscript by Jun 04 2022 11:59PM. If you will need more time than this to complete your revisions, please reply to this message or contact the journal office at plosone@plos.org. Please include the following items when submitting your revised manuscript:A rebuttal letter that responds to each point raised by the academic editor and reviewer(s). You should upload this letter as a separate file labeled 'Response to Reviewers'.A marked-up copy of your manuscript that highlights changes made to the original version. You should upload this as a separate file labeled 'Revised Manuscript with Track Changes'.An unmarked version of your revised paper without tracked changes. You should upload this as a separate file labeled 'Manuscript'.

We look forward to receiving your revised manuscript.

Kind regards,

Michael Bader

Academic Editor

PLOS ONE

Journal Requirements:

2. As part of your revision, please complete and submit a copy of the Full ARRIVE 2.0 Guidelines checklist, a document that aims to improve experimental reporting and reproducibility of animal studies for purposes of post-publication data analysis and reproducibility: https://arriveguidelines.org/sites/arrive/files/Author%20Checklist%20-%20Full.pdf (PDF). Please include your completed checklist as a Supporting Information file. Note that if your paper is accepted for publication, this checklist will be published as part of your article.

Reviewers' comments:

Reviewer's Responses to Questions

**Comments to the Author**

1. Is the manuscript technically sound, and do the data support the conclusions?

Reviewer #1: Yes

Reviewer #2: Partly

2. Has the statistical analysis been performed appropriately and rigorously? 

Reviewer #1: Yes

Reviewer #2: Yes

3. Have the authors made all data underlying the findings in their manuscript fully available?

Reviewer #1: Yes

Reviewer #2: No

4. Is the manuscript presented in an intelligible fashion and written in standard English?

Reviewer #1: Yes

Reviewer #2: Yes

5. Review Comments to the Author

Reviewer #1: In this manuscript, Teng et al. explore the relationships between oxygen exposure, oxidative stress, myeloperioxidase and endoplasmic reticulum (ER) stress in the context of oxygen exposure in prematurity and the development of bronchopulmonary dysplasia. MPO, OS and ER stress are all relevant aspects to hyperoxia induced lung injury and development of BPD but whether they are associative or interlinked is not known, and is tested here using S-D rat pups that are exposed to the ER stress inducers tunicamycin or to hyperoxia, along with retrospective examination of lung sections from patients with/without BPD. The authors find tunicamycin directly induces lung simplification and increases ER stress and severity of BPD. Inhibition of ER stress with the chaperone TUDCA or KYC (to inhibit MPO) showed a causal role for ER stress in BPD (that also reduces MPO) while inhibition of MPO reducing ER stress showed another level of causality and linkage. The authors conclude that inhibition of ER stress per se, regardless of its upstream regulation or origin is beneficial in the context of BPD, and a future therapeutic angle.

This is a very well done study with careful attention to rigor and reproducibility. The rationale for exploring the stated links is very strong, and there is substantial new data demonstrating causality in the rat model. The experiments use complementary techniques, and the results are easy to interpret. I have only a few minor concerns.

1. The authors need to re-examine the figures for neatness. Some of the text are overlapping with images, and there are some font oddities in some of the figures.

2. There should be some reference to how the perinatal and neonatal rat lung relates to the human lung developmental stages to show relevance of the hyperoxia rat model and the age groups used to the human condition

3. While the data are clear to someone familiar with the field, for readabiity, it is critical that an illustrative schematic showing the pathways and links is included as a final figure and/or a graphical abstract.

Reviewer #2: Dear Professor Dr. Bader,

I would like to thank you for the opportunity to review the submitted manuscript titled: “Role of endoplasmic reticulum stress in impaired neonatal lung growth and bronchopulmonary dysplasia”.

I found this article interesting and complicated to read. I can summarize the manuscript as follows:

Extreme premature neonates due to their immaturity often require mechanical ventilatory support and hyperoxic exposure to maintain life. They can develop as a complication to these exposures, Bronchopulmonary Dysplasia, which is a grave condition characterized phenotypically by alveolar simplification and vascular rarefaction. Excessive oxidative stress with inadequate anti-oxidant defenses and immature inflammatory responses are believed to be the main pathophysiological drivers to the establishment of the condition. The authors of this manuscript attempt to provide -without a clear hypothesis - a connection between oxidative and inflammatory pathways via endoplasmic reticulum stress to the development of BPD with the employment of both a human and an animal model.

In their experimental design, they first assess ESR stress in human lung autopsy specimens of children with the diagnosis of BPD vs controls.

They proceed with their animal experimental design with the independent application -in parallel - of two stressors. One is with hyperoxic exposure and the other is via a known pharmacological inducer of ERS which is tunicamycin. With their experiments the authors are able to show that both stressors induce ERS and apoptosis leading to lung phenotypes that include decreased alveolarization, vascularization and increased inflammatory cell infiltration. After the stressor application, they followed their experiments by providing pharmacological agents that serve as potential therapeutics that target the endoplasmic reticulum stress, providing information to prove that these agents can rescue from the pathologic phenotype.

In detail, newborn rats were exposed to hyperoxia in an established experimental model of Bronchopulmonary Dysplasia. The authors were able to elucidate lung morphologies, phenotypically similar in appearance with histopathological findings of bronchopulmonary dysplasia, with diminished alveolarization, decreased vascularization and increased inflammatory cell infiltration. Survival was also found decreased. With the employment of various techniques, they provide ample data to suggest that Hyperoxic exposure does lead to ERS which is attenuated via TUDCA (Tauroursodeoxycholic acid, an ER chaperone that can alleviate ERS) and KYC (N-acetyl83 lysyltyrosylcysteine amide, a proposed anti-inflammatory agent) administration.

The authors were able to show similar results with their parallel animal model using tunicamycin as the stressor. Tunicamycin decreased survival, weight gain and induced a lung phenotype similar to the previous model. The effects seem to be mitigated through TUDCA and KYC administration as well. Even though the experiments are in parallel and reveal similar outcomes it led the authors to conclude that the ERS response is fundamental for the development of BPD.

This conclusion is by association as no concomitant exposure to both stressors was utilized nor the exact techniques were employed between the parallel models. The authors rely heavily on their previous work which these experiments supplement. The novelty of this paper is the use of these potential therapeutic agents in the same experimental design, however they are reported and compared individually. Additional critical details on the methods need to be communicated.

Authors display the following information:

Survival: Yes

Population: Partially

Weight: Yes

Sex: No

Experimental Design Graph: No

Limitations: No

Major concerns

1. It is important to state a clear hypothesis of this project.

2. In the manuscript we are able to see the number of human subjects included in the study. However, it is unclear how many belong to the BPD group and how many are the controls. The age range is very wide and the cause of demise of the controls is not clear. The number of human subjects with BPD and without needs to be clearly stated as a BPD subject may be paired to multiple controls.

3. Murine hyperoxic models of bronchopulmonary dysplasia are commonly utilized. In a paragraph, it is imperative to note why the authors utilized such a model and the correlation with human disease and lung development.

4. Sex should be considered as a biological variable and data should indicate numbers of males and females

5. Tunicamycin seems to create a lung phenotype resembling the hyperoxic model of bronchopulmonary dysplasia. However, the authors utilize “rat BPD” group as seen in Figure 1. Is “rat BPD” group referring to the hyperoxic model of BPD, Tunicamycin or both? Authors should be careful when interchangeably use BPD in animal designations. Words such as “BPD phenotype” or “Hyperoxic” or any other stressor are preferred so that they do not create confusion.

6. The authors touch upon mitochondrial interactions and report on mitochondrial fission as evidenced by decreased expressions of two major Mitochondrial Associated Membrane proteins and mitochondrial fragmentation. This is mainly displayed in their tunicamycin model and indirectly reported for the hyperoxia model. What were the hyperoxia results of such finding compared to normoxia?

7. Ideally all 4 groups should be displayed in Fig8 (21%O2-PBS, 21%O2-TUDCA,90%O2-PBS, 90%O2- TUDCA)

8. Since the authors state that multiple pharmacological agents may be important to combat BPD it would be novel if they administered both TUDCA and KYC and assess results in their animal Hyperoxic model of BPD. The tunicamycin model will supplement the abovementioned experiment. Groups could be 21%O2-PBS, 21%O2-TUDCA, 21%O2-KYC, 90%O2-PBS, 90%O2- TUDCA, 90%O2- KYC, 90%O2-TUDCA-KYC)

9. Fig4 displays 21%O2-DMSO vs 21%O2-Tunicamycin. This implies experimentation in normoxia as clearly stated in the manuscript. Was Hyperoxia ever used in conjunction to tunicamycin in the experiments?

10.There are numerous limitations of the model. The authors should recognize in a paragraph what these limitations are and why their model of disease is the best to prove their hypothesis.

11. A graphical abstract or graph explaining the experimental design would be greatly helpful.

Minor Concerns

1. Fig 3 needs scale bars; Overlapping graphs and missing “Tunicamycin” Fig 3A.

2. Fig 4A missing “Tunicamycin” ; Fig 4D has a zoomed section; No percentage of zoom is displayed

3. Fig 5A Bold font on bar graph

4. Fig6 Overlapping graph titles & inconsistent use of group names (Tunicamycin+PBS vs Tunicamycin-PBS, etc)

5. Fig 8A TUDCA name in bar needs to be corrected.

6. Inconsistent use of abbreviations. Hyperoxia (HOX) is changed in some graphs as “>90%O2” while in others is “HOX” while in others is suspected “ratBPD”.

6. PLOS authors have the option to publish the peer review history of their article (what does this mean?). If published, this will include your full peer review and any attached files.

Reviewer #1: No

Reviewer #2: No

---

## [Author Response · Author response to Decision Letter 0]

11 May 2022

Reviewer #1:

This is a very well-done study with careful attention to rigor and reproducibility. The rationale for exploring the stated links is very strong, and there is substantial new data demonstrating causality in the rat model. The experiments use complementary techniques, and the results are easy to interpret. I have only a few minor concerns. 

1. The authors need to re-examine the figures for neatness. Some of the text are overlapping with images, and there are some font oddities in some of the figures. 

We thank the reviewer for bringing this issue to our attention. The problem we encountered at converting ppt into pdf is resolved. We have gone through all the figures to correct inconsistencies, font, and format problems. We have enclosed a corrected set of figures.

2. There should be some reference to how the perinatal and neonatal rat lung relates to the human lung developmental stages to show relevance of the hyperoxia rat model and the age groups used to the human condition. 

We agree that the essential information is important for general readers and have added it in the introduction with a relevant reference (lines 70-82). 

3. While the data are clear to someone familiar with the field, for readability, it is critical that an illustrative schematic showing the pathways and links is included as a final figure and/or a graphical abstract. 

Thank you for this great suggestion. We have prepared a figure (Figure 11) to summarize the findings in our manuscript and connected these new data with our previously reported “cycle of destruction”. The figure can be found in the conclusion section (line 570). 

Reviewer #2

Major concerns 

1. It is important to state a clear hypothesis of this project. 

We have added our hypothesis into the introduction (lines 93-95).

2. In the manuscript we are able to see the number of human subjects included in the study. However, it is unclear how many belong to the BPD group and how many are the controls. The age range is very wide and the cause of demise of the controls is not clear. The number of human subjects with BPD and without needs to be clearly stated as a BPD subject may be paired to multiple controls. 

We apologize for not clearly stating the 1:1 match, 10 specimens in each group (line 150 and lines 275-276), for the age- and sex-matched human study. We have added a supplemental Table S1 including their age of death, sex, and primary diagnosis. 

3. Murine hyperoxic models of bronchopulmonary dysplasia are commonly utilized. In a paragraph, it is imperative to note why the authors utilized such a model and the correlation with human disease and lung development. 

Thanks for pointing this out. We have added the reason why the rat hyperoxic model was chosen in this study (lines 70-82). As rats are born with a well-developed antioxidant system, we have to use a high concentration of oxygen to elicit lung damage. The pros and cons of using the murine model are added to the revised introduction.

4. Sex should be considered as a biological variable and data should indicate numbers of 

males and females. 

Thanks for pointing out the importance of biological variables, especially sex, to the results. We randomly reallocated all pups by sex into 2 or 4 nursing dams to minimize the sex effect in our experiments. The small number of pups in each specific experiment prevented the detection of a statistical difference between the sexes. We thus determined to use data from both sexes for comparisons in each experiment. The number of each sex was added to the results as suggested. 

5. Tunicamycin seems to create a lung phenotype resembling the hyperoxic model of bronchopulmonary dysplasia. However, the authors utilize “rat BPD” group as seen in Figure 1. Is “rat BPD” group referring to the hyperoxic model of BPD, Tunicamycin or both? Authors should be careful when interchangeably use BPD in animal designations. Words such as “BPD phenotype” or “Hyperoxic” or any other stressor are preferred so that they do not create confusion. 

We appreciate the invaluable suggestion to use different descriptors in different situations to avoid confusion for the readers. We have carefully gone over the manuscript to make sure that HOX rat lung and Tun-treated rat lung are used for each condition. 

6. The authors touch upon mitochondrial interactions and report on mitochondrial fission as evidenced by decreased expressions of two major Mitochondrial Associated Membrane proteins and mitochondrial fragmentation. This is mainly displayed in their tunicamycin model and indirectly reported for the hyperoxia model. What were the hyperoxia results of such finding compared to normoxia? 

We previously reported the increased DRP expression in HOX rat lungs as compared to NOX rat lungs (Teng RJ et al. Attenuation of endoplasmic reticulum stress by caffeine ameliorates hyperoxia-induced lung injury. Am J Physiol Lung Cell Mol Physiol. 2017 May 1;312(5):L586-L598.) but did not repeat the same experiment in this study. The information has been added to the results (lines 336-337) and discussion (lines 442-445). We have made the correction in the discussion and referenced that report accordingly. 

7. Ideally all 4 groups should be displayed in Fig8 (21%O2-PBS, 21%O2-TUDCA,90%O2-PBS, 90%O2- TUDCA) 

The 4-group study was performed to compare weight gain, survival rate, and morphometric analyses (Figure 9) but we did not do so for the proteomic studies in Figure 8. However, the results in Figure 9 support the safety of TUDCA in protecting the rat pups under HOX. 

8. Since the authors state that multiple pharmacological agents may be important to combat BPD it would be novel if they administered both TUDCA and KYC and assess results in their animal Hyperoxic model of BPD. The tunicamycin model will supplement the abovementioned experiment. Groups could be 21%O2-PBS, 21%O2-TUDCA, 21%O2-KYC, 90%O2-PBS, 90%O2- TUDCA, 90%O2- KYC, 90%O2-TUDCA-KYC) 

We deeply appreciate your suggestion and will incorporate into our future studies. Such an experiment with 7 treatment groups will definitely provide important information about treatment strategy and the synergistic/additive effect of two promising chemicals in attenuating BPD. We estimate that such an experiment will require at least 7 dams, and roughly 70 pups, for randomization at the same time which is beyond the capacity of our current animal hyperoxia chamber and space for animals. Our 4-group experiments usually take 2-3 months to accomplish. To pursue the 7-group study, we expect at least 3-5 months will be required. The gestation for SD rats ranges between 21 and 23 days, so the delivery can be 3 days apart between dams which further complicates the randomization. Your suggestion is truly a great idea and will be our future research direction. 

9. Fig4 displays 21%O2-DMSO vs 21%O2-Tunicamycin. This implies experimentation in normoxia as clearly stated in the manuscript. Was Hyperoxia ever used in conjunction to tunicamycin in the experiments? 

Our preliminary experiment showed that rat pups were sensitive to tunicamycin with one injection in the NOX group leading to a relatively high death rate (~40%) by P7. Considering that 20-25% of HOX pups died in our previous experiments, we estimated that a combination of HOX and tunicamycin will lead to a ~50% death rate. We, therefore, did not attempt to use the combined HOX-Tun treatment to study the efficacy of TUDCA. Additionally, we observed the protective effect of TUDCA for the impaired alveolar formation in HOX-alone and Tun-alone treated rat lungs and concluded that these findings provide enough evidence to support the role of ER stress in BPD. 

10. There are numerous limitations of the model. The authors should recognize in a paragraph what these limitations are and why their model of disease is the best to prove their hypothesis. 

We agree that animal models of BPD have limitations. The rat hyperoxia model has been used for decades for studying BPD. The important concerns are the mature antioxidant system in newly born rat pups, unlike the premature infants, and the use of hyperoxia as single stress while BPD in premature infants may have a multi-factorial origin. We addressed these limitations in the section of the revised introduction (lines 70-82). 

11. A graphical abstract or graph explaining the experimental design would be greatly helpful. 

Thank you for the advice. The other reviewer also had a similar suggestion. We created a new figure, figure S1, in our revised manuscript (lines 160-161). 

Minor Concerns – 

1. Fig 3 needs scale bars; Overlapping graphs and missing “Tunicamycin” Fig 3A. 

2. Fig 4A missing “Tunicamycin”; Fig 4D has a zoomed section; No percentage of zoom is displayed 

3. Fig 5A Bold font on bar graph 

4. Fig6 Overlapping graph titles & inconsistent use of group names (Tunicamycin+PBS vs Tunicamycin-PBS, etc) 

5. Fig 8A TUDCA name in bar needs to be corrected. 

We apologize for not realizing that transforming ppt files into pdf files created so many mistakes. We have carefully made corrections for each one of them in the resubmission. 

6. Inconsistent use of abbreviations. Hyperoxia (HOX) is changed in some graphs as “>90%O2” while in others is “HOX” while in others is suspected “rat BPD”. 

Thanks for pointing them out. We have gone over all our figures to make those abbreviations consistent.

---

## [Decision Letter · Decision Letter 1]

24 May 2022

Role of endoplasmic reticulum stress in impaired neonatal lung growth and bronchopulmonary dysplasia

PONE-D-22-06017R1

Dear Dr. Teng,

We’re pleased to inform you that your manuscript has been judged scientifically suitable for publication and will be formally accepted for publication once it meets all outstanding technical requirements.

Kind regards,

Michael Bader

Academic Editor

PLOS ONE

Additional Editor Comments (optional):

Reviewers' comments:

Reviewer's Responses to Questions

**Comments to the Author**

1. If the authors have adequately addressed your comments raised in a previous round of review and you feel that this manuscript is now acceptable for publication, you may indicate that here to bypass the “Comments to the Author” section, enter your conflict of interest statement in the “Confidential to Editor” section, and submit your "Accept" recommendation.

Reviewer #1: All comments have been addressed

Reviewer #2: All comments have been addressed

2. Is the manuscript technically sound, and do the data support the conclusions?

Reviewer #1: Yes

Reviewer #2: Yes

3. Has the statistical analysis been performed appropriately and rigorously? 

Reviewer #1: Yes

Reviewer #2: Yes

4. Have the authors made all data underlying the findings in their manuscript fully available?

Reviewer #1: Yes

Reviewer #2: Yes

5. Is the manuscript presented in an intelligible fashion and written in standard English?

Reviewer #1: Yes

Reviewer #2: Yes

6. Review Comments to the Author

Reviewer #1: (No Response)

Reviewer #2: Dear Professor Dr. Bader,

The authors of this manuscript have addressed both reviewers’ comments and concerns according to the scope of the experimental limitations. As mentioned previously, the authors have completed an array of experiments utilizing two different stressors. These were done in parallel, in order to demonstrate ,via association, that inflammation and oxidative stress lead to the development of Bronchopulmonary Dysplasia by inducing endoplasmic reticulum stress. These findings give an insight to our understanding of the common pathways that may be utilized for the development of this grave condition.

The authors have updated the manuscript to display the following information:

Survival: Yes

Population: Yes

Weight: Yes

Sex: Yes

Experimental Design Graph: Yes

Limitations: Yes

To conclude, I recommend the acceptance for publication of this manuscript.

All the best.

7. PLOS authors have the option to publish the peer review history of their article (what does this mean?). If published, this will include your full peer review and any attached files.

Reviewer #1: No

Reviewer #2: No

---

## [Editor Report · Acceptance letter]

17 Aug 2022

PONE-D-22-06017R1 

Role of endoplasmic reticulum stress in impaired neonatal lung growth and bronchopulmonary dysplasia 

Dear Dr. Teng:

I'm pleased to inform you that your manuscript has been deemed suitable for publication in PLOS ONE. Congratulations! Your manuscript is now with our production department. 

Kind regards, 

on behalf of

Prof. Michael Bader 

Academic Editor

PLOS ONE